

# The FuGas 2.1 framework for atmosphere-ocean coupling in geoscientific models: improving estimates of the solubilities and fluxes of greenhouse gases and aerosols

Vasco M. N. C. S. Vieira[1], Pavel Jurus[2,5], Emanuela Clementi[3], Heidi Pettersson[4] and Marcos Mateus[1].

[1]MARETEC, Instituto Superior Técnico, Universidade de Lisboa, Av Rovisco Pais, 1049-001 Lisboa, Portugal.

[2]DataCastor, U Svobodarny 1063/6,190 00 Praha 9, Prague, Czech Republic.

[3]Istituto Nazionale di Geofisica e Vulcanologia, INGV, Bologna, Italy.

[4]Finnish Meteorological Institute, P.O. Box 503, FI-00101 Helsinki, Finland.

[5]Institute of Computer Science, Czech Academy of Sciences, Prague, Czech Republic.

*Correspondence to*: Vasco M. N. C. S. Vieira vasco.vieira@tecnico.ulisboa.pt

## ABSTRACT

Accurate estimates of the atmosphere-ocean balances and fluxes of greenhouse gases and aerosols are fundamental for geoscientific models dealing with climate change. A significant part of these fluxes occur at the coastal ocean which, although much smaller than the open ocean, is also much more heterogenic. The scientific community is becoming increasingly aware of the necessity to model the Earth at finer spatial and temporal resolutions, which also requires better descriptions of the chemical, physical and biological processes involved. The standard formulations for the gas

transfer velocities and solubilities are 24 and 36 years old, respectively, and recently, new alternatives have emerged. We developed a framework congregating the geophysical processes involved which are customizable with alternative formulations with different degrees of complexity and/or different theoretical backgrounds. We propose this framework as basis for novel couplers of atmospheric and oceanographic model components. We tested it with fine resolution data from the European coastal ocean. Although the benchmark and alternative solubility formulations agreed well, their

minor divergences yielded differences of many tons of greenhouse gases dissolved at the ocean surface. The transfer velocities largely mismatched their estimates, a consequence of the benchmark formulation not considering factors that were proved determinant at the coastal ocean. Climate Change research requires more comprehensive simulations of atmosphere-ocean interactions but the formulations able to do it require further calibration and validation.


Keywords: solubility, transfer velocity, Henry constant.



## 1    Introduction

Earth-System as well as Regional models are ensembles of inter-connected components, namely the land, ocean,
atmosphere and cryosphere. The exchange of information between each pair requires specific couplers that are also
responsible for the estimation of geophysical processes specific to their physical interfaces. In this work we focus on the
coupling between the atmospheric and the oceanographic components, and the estimation of the air-water fluxes of
greenhouse gases and aerosols. The Code and Data Availability section has the link to the software, data and videos.

Because the oceans can act as sinks or sources of greenhouse gases and aerosols to the atmosphere, the
dynamics of their gas exchanges are fundamental for Earth's climate. The open ocean is generally believed to uptake
$CO_2$ from the atmosphere, despite the observed seasonal, inter-annual and regional variability. In the pole regions the
solubility pump retrieves large amounts of greenhouse gases from the atmosphere and transports them to the deep
ocean. On the other hand, the balances and fluxes of $CO_2$, $CH_4$, $N_2O$ and DMS at the coastal oceans' surface are very
heterogenic due to factors like upwelling, plankton productivity and continental loads. Earth-System Models (ESM) and
marine biogeochemistry usually simulate the biosphere at decadal and centennial time-scales with daily intervals and
spatial resolutions of hundreds to one thousand kilometres (see  ESM applications by IPCC, MPI or CMCC).
Constrained by calculus demands, they estimate the atmosphere-ocean gas fluxes from simpler formulations that
disregard the complexity of processes more recently unveiled at the coastal ocean. The generalization by Wanninkhof
(1992), relying on wind speed ($u_{10}$) as the sole driver of transfer velocity, is the standard in current ESM at coarse
resolutions. Alternatively, the regional oceanographic numerical lab MOHID allows the user to choose between the air-
water gas exchange formulations by Carini et al. (1996) and Raymond and Cole (2001), only accounting for $u_{10}$, or by
Borges et al. (2004), also accounting for current drag with the bottom. These are empirical formulations best fitting low
wind data collected from estuaries. There are many other simpler formulations estimating the air-water gas exchange
considering a few factors that were determinant for that specific set of environmental conditions and optimizing the
adjustment to their specific data. But modelling the coastal oceans with fine resolution requires an algorithm that,
whatever the local conditions, is always able to forecast with improved accuracy due to its enhanced representation of
the multitude of processes potentially present. Developing such algorithm demands for a framework able to be updated
with the best formulation for each of the mediator processes involved.

The FuGas 2.1 is an upgrade of the framework by Vieira et al. (2013) congregating several of the geophysical
processes involved in the air-water gas exchanges, and where each process can be simulated by one formulation chosen
from an extensive list. It includes 50 alternative formulations to account for such factors as solubility, wind or current
mediated turbulence, atmospheric stability, sea-surface roughness, breaking waves, air and water viscosities,
temperature and salinity. We use the FuGas 2.1 to compare between the estimations of the solubilities and fluxes of
greenhouse gases using the ESM standards and recent alternative formulations. First, we tested with field data from the
Baltic Sea. Then, we coupled the Weather Research and Forecasting (WRF) atmospheric model to the WaveWatch III
(WW3) - NEMO oceanographic model using simulated data from the European coastal ocean. The calculus was
vectorized and parallelized for improved computational speed.

## 2    Methods

Air-water gas fluxes result from the interaction of two factors: (i) the unbalance between the gas concentrations in the
air and in the water sets the strength and direction of the flux, and (ii) the resistance the medium does for being crossed



by the flow. The traditional formulation estimates the flux from $F=k_w \cdot k_H cp \cdot \Delta p_{gas}$, in units of $mol \cdot m^{-2} \cdot s^{-1}$. The $\Delta p_{gas}$ is the difference between air and water gas partial pressures (atm). The $k_H cp$ is the Henry's constant for the gas solubility in its $C_w/p_a$ form ($mol \cdot m^{-3} \cdot atm^{-1}$), where $p_a$ is its air partial pressure (atm) and $C_w$ its concentration in the water ($mol \cdot m^{-3}$).

The $k_w$ is the transfer velocity of gases across the sub-millimetrically thick water surface layer in $m \cdot s^{-1}$ although usually plotted in $cm \cdot h^{-1}$. The alternative double layer model (Liss and Slater, 1974) estimates the flux taking into consideration both the water-side and air-side sub-millimetrically thick surface layers and thus, $F = K_w(C_a/k_H - C_w) = K_a(C_a - C_w \cdot k_H)$. The $C_a$ and $C_w$ are the concentrations of the gas in air and water given in $mol \cdot m^{-3}$ and the $k_H$ is Henry's constant in its equivalent dimensionless quantity ($C_a/C_w$). The transfer velocity is averaged over both layers from $K_w=(1/k_w+1/(k_H \cdot k_a))^{-1}$

or its equivalent $K_a=(k_H/k_w+1/k_a)^{-1}$.

## 2.1 Solubility

Sarmiento and Gruber (2013) compiled the algorithm for the $k_H cp$ dependence on temperature and salinity provided by Weiss (1974) and Weiss and Price (1980). We converted it to its corresponding dimensionless $k_H$ preserving the

constants required to estimate Bunsen's solubility coefficient β. This formulation accounted for fugacity (f) of non-ideal gases (Eq. 1) and corrected the gas partial pressure for moisture effects from the expression $p_{moist}=(1-p_{H2O}/P)p_{dry}$ considering water vapour saturation over the sea-surface (Eq. 2). P is air pressure (atm), $T_w$ is water temperature (K), S is salinity (‰), p is the gas partial pressure (atm), R is the ideal gas law constant ($Pa \cdot m^3 \cdot mol^{-1} \cdot K^{-1}$), $V_m$ is the molar volume of the specific gas (22.3 for $CO_2$ and $CH_4$, and 22.2432 for $N_2O$) and $V_{ideal}=22.4136 \ mol \cdot L^{-1}$ is the molar

volume of ideal gases. Solubility coefficients were estimated from the Virial expansion (Eq. 3), where B was β or $\beta/V_m$, depending on which gas it was applied to (Table 3.2.2 in Sarmiento and Gruber (2013)). Our software automatically detected the gas from the $a_i$ coefficient. When B=β the $k_H$ was estimated from Eq. (4). When $B=\beta/V_m$ the $k_H$ was estimated from Eq. (5).

$$f = \exp\left(\frac{101.325P(V_m - V_{ideal})}{RT_w}\right) \tag{1}$$

$$\log\frac{p_{H_2O}}{P} = 24.4543 - 67.4509\left(\frac{100}{T_w}\right) - 4.8489\ln\left(\frac{T_w}{100}\right) - 0.000544S \tag{2}$$

$$\log(B) = a_1 + a_2\frac{100}{T_w} + a_3\log\frac{T_w}{100} + a_4\left(\frac{T_w}{100}\right)^2$$
$$+ S \cdot \left(b_1 + b_2\frac{T_w}{100} + b_3\left(\frac{T_w}{100}\right)^2\right) \tag{3}$$

$$k_H = \left(1 - \frac{p_{H_2O}}{P}\right)\frac{101.325V_m}{RT_w\beta f} \tag{4}$$

$$k_H = \frac{101.325}{RT_w\beta f} \tag{5}$$


Johnson (2010) developed an algorithm from an alternative chemistry background. It accounts for the effects of temperature and salinity taking into consideration the molecular and thermodynamic properties of the water, its solutes and the specified gas, but disregarding the non-ideal behaviour of the gases and moisture. His formulation was developed from the compilation by Sander (2015) (although available in the web since 1999) of the $k_H cp$ for nearly all

gases in the atmosphere at 25º C (298.15 K) and 0 ppt. Then, equation (6) converted the $k_H cp$ to $k_H$ at a given temperature and 0 ppt salinity. The term $-\Delta_{soln}H/R$ reflected the temperature (in Kelvin) dependence of solubility, having a value of 2400 for $CO_2$, 1700 for $CH_4$ and 2600 for $N_2O$. The correction to a given salinity (Eq. 7) relied on the



empirical Setschenow constants ($K_S = \theta \cdot \log Vb$) reporting the effect of electrolytes salting-out gases proportionally to their liquid molar volume at boiling point (Vb). The Vb was estimated using the additive Schroeder method, whereas $\theta$

was estimated from Eq.8 using a provisional $k_{H\#} = 0.0409/k_H cp$.

$$k_{H,0} = \frac{12.1866}{P \cdot T_w \cdot k_{H,cp} \cdot e^{\frac{-\Delta_{Soln}H}{R}(1T_w - 1298.15)}} \tag{6}$$

$$k_H = k_{H,0} \cdot 10^{K_S S} \tag{7}$$

$$
\begin{aligned}
\theta = \; & 7.33532 \cdot 10^{-4} + 3.39615 \ast 10^{-5} \cdot \log(k_{H\#}) \\
& -2.40888 \cdot 10^{-6} \cdot \log(k_{H\#})^2 \\
& +1.57114 \cdot 10^{-7} \cdot \log(k_{H\#})^3
\end{aligned}
\tag{8}
$$


## 2.2     Transfer velocity

The available algorithms consider that the rate at which gases cross the sea-surface is basically set by the turbulence upon it. E.g. wind drag, wave breaking, currents and rain promote turbulence. The water viscosity, set by temperature and salinity and enhanced by the presence of surfactants, antagonizes turbulence. With all these forcings, it becomes

difficult to develop an algorithm that estimates the transfer velocity accurately. The literature has many of them, either fitted to specific surface conditions or rougher generalizations, focusing on different factors and relying in different theoretical backgrounds. The simpler ones rely on the wind velocity 10m above the sea-surface ($u_{10}$). Among then, the formulation by Wanninkhof (1992) (henceforth also mentioned as 'Wan92') became the standard used in ESM and satellite data processing. It further considers the Schmidt number of the water ($Sc_w$) related to viscosity and with its

exponent reflecting the surface layer's rate of turbulent renewal, and the temperature dependent chemical enhancement due to $CO_2$ reaction with water ($\alpha_{Ch}$):

$$k_w = (\alpha_{Ch} + 0.31 \cdot u_{10}^2)\left(\frac{Sc_w}{660}\right)^{-0.5} \tag{9a}$$

$$\alpha_{Ch} = 2.5 \cdot (0.5246 + 0.0162T_w + 0.000499T_w^2) \tag{9b}$$


Other simple empirical formulations based only on $u_{10}$ (Carini et al., 1996; Raymond and Cole, 2001), or also accounting for current drag with the bottom (Borges et al., 2004), used data collected in estuaries under low wind conditions. However, modelling the coastal ocean at finer resolutions requires an enhanced representation of the multitude of processes involved. Hence, we updated the framework by Vieira et al. (2013), with the $k_w$ being

decomposed into its shear produced turbulence ($k_{wind}$) and bubbles from whitecapping ($k_{bubble}$) forcings (Asher and Farley, 1995; Borges et al, 2004; Woolf, 2005; Zhang et al., 2006). The effect of currents was disregarded at this stage (Eq. 10). $Sc_w$ was determined from temperature and salinity following Johnson (2010).

$$k_w = (\alpha_{Ch} + k_{bubble} + k_{wind}) \cdot (600/Sc_w)^{0.5} \tag{10}$$


The formulation by Zhao et al. (2003), merged $k_{wind}$ into $k_{bubble}$ (Eq. 11a) using the wave breaking parameter ($R_B$ given by Eq. 11b). The $u_*$ is the friction velocity i.e, the velocity of wind dragging on the sea-surface, and $f_p$ is the peak angular frequency of the wind-waves. The kinematic viscosity of air ($\upsilon_a$) was estimated from Johnson (2010). This





solution used the wave field as a proxy for whitecapping that increased transfer velocity with wind-wave age. However,
it simultaneously used the wave field as a proxy for the sea-surface roughness that increased transfer velocity from
wind-drag over steeper younger waves (through the WLLP estimation of $u_*$ explained in a section below).

$$k_{bubble} = 0.1315 \cdot R_B^{0.6322} \tag{11a}$$

$$R_B = \frac{u_*^2}{2\pi f_p \nu_a} \tag{11b}$$


A more comprehensive solution split the two drives of transfer velocity (Woolf, 2005; Zhang et al., 2006): $k_{wind}$ for the
transfer mediated by the turbulence generated by wind drag (Eq. 12) (Jähne et al., 1987) and $k_{bubble}$ for the transfer
mediated by the bubbles generated by breaking waves (Eq. 13) (Zhang et al., 2006). B is Bunsen's solubility coefficient
estimated for the local sea-surface conditions. $W=3.88\times10^{-7}R_B^{1.09}$ is the whitecap cover requiring the $R_B$ estimated from
(Eq. 11b), V=4900, e=14 and n=1.2.

$$k_{wind} = 1.57 \cdot 10^{-4} \cdot u_* \tag{12}$$

$$k_{bubble} = \frac{WV}{B}[1 + (e \cdot B \cdot Sc_w^{-12})^{-1n}]^{-n} \tag{13}$$

These formulations required friction velocity ($u_*$), which was estimated from the Wind Log-Linear Profile (WLLP: Eq.
14) accounting for wind speed at height z ($u_z$), atmospheric stability of the surface boundary layer (through $\psi_m$) and sea-
surface roughness (through the roughness length $z_0$). The $\kappa$ is von Kármàn's constant.

$$u_* = \frac{u_z \cdot \kappa}{\ln(z) - \ln(z_0) + \psi_m(z, z_0, L)} \tag{14}$$


Roughness length ($z_0$) is the theoretical minimal height (most often sub-millimetrical) at which wind speed averages
zero. It is dependent on surface roughness and often used as its index. It is more difficult to determine over water than
over land as there is a strong bidirectional interaction between wind and sea-surface roughness. Taylor and Yelland
(2001) proposed a dimensionless $z_0$ dependency from the wave field, increasing with the wave slope (Eq. 15). Due to
the bidirectional nature of the $z_0$ and $u_*$ relation, we also tested an iterative solution (iWLP) where Eq.15 was used as a
first guess for the $z_0$ and Eq.14 for its subsequent $u_*$. A second iteration re-estimated $z_0$ from the COARE 3.0 (Fairall et
al.; 2003) adaptation of the Taylor and Yelland (2001) formulation, which added a term for smooth flow (Eq. 16), and
$u_*$ again from Eq.14. Applying four iterations were enough for an excellent convergence of the full data array.

$$\frac{z_0}{H_s} = 1200 \cdot \left(H_s L_p\right)^{4.5} \tag{15}$$

$$z_0 = 1200 \cdot H_s \left(\frac{H_s}{L_p}\right)^{4.5} + \frac{0.11\nu_a}{u_*} \tag{16}$$

Atmospheric stability characterized the tendency of the surface boundary layer (SBL) to be well mixed (unstable SBL
with $\psi_m<0$) or stratified (stable SBL with $\psi_m>0$). It was inferred from the 'bulk Richardson number' ($Ri_b$: Eq. 17),
weighting the air vertical heat gradient and kinetic energy. Its estimation required the air virtual potential temperature,





in its turn estimated from air temperature, air pressure and specific humidity (Grachev and Fairall, 1997) or from the liquid water mixing ratio (Stull, 1988). Alternatively, the use of the air potential temperature neglected humidity (Lee, 1997). The wind velocity ($u_z$), temperature ($T_z$), pressure ($P_z$) and humidity ($q_z$) z meters above sea-surface were given by the WRF second level. The wind velocity at $z_0$ ($u_0$) was set to the theoretical $u_0=0$. Temperature at the height of 0 m

($T_0$) was given by the SST (Grachev and Fairall, 1997; Fairal et al., 2003; Brunke et al., 2008) without rectification for cool-skin and warm-layer effects due to the lack of some required variables. Yet, these effects tend to compensate each other (Brunke et al., 2008; Fairall et al., 1996; Zeng and Beljars, 2005). Air pressure at 0 m ($P_0$) was given by the WRF at the lower first level (at roughly 0 m). Humidity at 0 m ($q_0$) was set to the saturation level at $P_0$ and $T_0$ (Grachev and Fairall, 1997). The $Ri_b$ was used to estimate the length L from, Monin-Obukhov's similarity theory, a discontinuous

exponential function tending to $\pm\infty$ when $Ri_b$ tends to $\pm 0$ and tending to $\pm 0$ when $Ri_b$ tends to $\pm\infty$. $Ri_b$ and L were used to estimate $\psi_m$ following Stull (1988) or Lee (1997) algorithms.

$$Ri_b = \frac{g \Delta T \Delta z_i}{T \cdot u_z^2} \qquad (17)$$

$CO_2$ is mildly soluble with a $K_H=1.17$ for pure water at 25 ºC. Its transfer velocity is limited by the molecular crossing of the water-side surface layer. $CH_4$ is much less soluble with a $K_H=31.5$ for pure water at 25 ºC. Its transfer velocity should also take into consideration the molecular crossing of the air-side surface layer (Johnson, 2010). We compared between the use of the traditional single layer and the double layer "thin film" model (Liss and Slater, 1974; Johnson, 2010; Vieira et al, 2013), the later requiring the air-side transfer velocity ($k_a$) estimated from the COARE formulation as

in Eq. 18 (Jeffrey et al., 2010). CD is the drag coefficient and $Sc_a$ the Schmidt number of air, which were determined for a given temperature and salinity following Johnson (2010).

$$k_a = \frac{u_*}{13.3 \cdot Sc_a^{12} + CD^{12} - 5 + \frac{\log(Sc_a)}{2\kappa}} \qquad (18)$$

**2.3    Validation with field data**

The field sampling occurred from the 22$^{nd}$ of May 2014 to the 26$^{th}$ of May 2014 using the atmospheric tower at Östergarnsholm in the Baltic Sea (57° 27′ N, 18° 59′ E), the Submersible Autonomous Moored Instrument (SAMI-$CO_2$) 1 km away and the Directional Waverider (DWR) 3.5 km away, both south-eastward from the tower (see e.g. Högström et al. (2008) and Rutgersson et al. (2008) for detailed description of the sites). The air-water $CO_2$ fluxes

measured by eddy-covariance were smoothed over 30 min bins and corrected according to the Webb-Pearman-Leuning (WPL) method (Webb et al., 1980). We used only the fluxes for which the wind direction set the SAMI-$CO_2$ and DWR in the footprint of the atmospheric tower (90º < wind direction < 180º). The DWR measured temperatures at 0.5 m depth, taken as representative for the sea-surface. Salinity was obtained from the Asko mooring data provided by the Baltic In-Situ Near-Real-Time Observations available in Copernicus Marine catalogue. We applied this data set to the

single processing software ensemble of the FuGas 2.1 in order to test which algorithms provide better approximations to reality.





### 2.4    Atmosphere-ocean coupler

The atmospheric model was the standard operational application of the WRF by Meteodata.cz , with 9 km and 1 h resolutions. Air temperature 'T' (ºC), pressure 'P' (atm), U and V components of wind velocity (m·s$^{-1}$), water vapour mixing ratio 'Q' (scalar) and height 'h' (m), where retrieved at the two lowest levels within the atmospheric surface boundary layer (SBL). The vertical thickness of the WRF horizontal layers varied with space and time. Over the ocean, the two lowest levels occurred roughly at 0 m and 12 m heights. The WRF output decomposes height, temperature and
pressure into their base level plus perturbation values.

Sea-surface temperature (SST) and salinity (S) were estimated by the NEMO modelling system provided in the MyOcean catalogue with 1/12º and 1 day resolutions. The WW3 wave field data for the Mediterranean Sea was supplied by INGV using the WW3-NEMO modelling system at 0.0625 º and 1 h resolutions (Clementi, 2013), and for the North Atlantic by Windguru at roughly 0.5º and 3 h resolutions. The variables included significant wave height 'H$_s$'
(m) and peak frequency 'f$_p$' (rad·s$^{-1}$) for wind sea i.e, disregarding swell.  A few aspects did not correspond to the ideal data format for atmosphere-ocean coupling, and required further calculations: (i) The peak wave length 'L$_p$' (m) was estimated from the peak frequency assuming the deep-water approximation: $L_p=2\pi g/f_p^2$, where g is the gravitational acceleration constant; (ii) the Windguru data did not provide wind sea component (where and) when the wind was too low. For these missing cases were attributed the lowest H$_s$ and L$_p$ simulated everywhere else; (iii) the Windguru and the
INGV data overlapped along the Iberian shores, in which case the INGV was given a 2:1 weight over the Windguru data.

The WRF and WW3-Nemo outputs were retrieved for the European shores from the 24$^{th}$ of May 2014 at 06h to the 27$^{th}$ of May 2014 at 00 h. All variables were interpolated to the same 0.09º grid (roughly 11 km at Europe's latitudes) and 1 h time steps. This resulted in a data set with 17 variables × 41776 locations × 66 time instances, that
occupied nearly 1Gb ram memory (with another 1Gb taken by the software). To optimize the computations, the calculus was first vectorized and then parallelized using the Single Program Multiple Data (spmd) programming strategy. Hence, in the FuGas 2.1 multiple processing software ensemble, the variables were first organized in matrices with locations along the 1$^{st}$ dimension and time along the 2$^{nd}$. Running the calculus applying matrix algebra to the whole data set, by itself represented an improved speed of several orders of magnitude. Furthermore, the spmd replicated the data, split the
replicates into n approximately equal-sized arrays, and distributed their calculus among the n available cpu cores, which represented an extra improvement of computational speed. However, it also bared computational costs: (i) invoking the parallel processing toolbox was time consuming, (ii) replicating 1Gb ram was time consuming, (iii) once running the calculus, the 4Gb ram memory was soon exhausted, which stall the calculus, (iv) to avoid it, the spmd were split into several sequential code blocks and in-between the variables no longer necessary were deleted. This spmd fragmentation
was time consuming. In conclusion, there is no perfect solution for calculus parallelization, and although spmd is the best strategy available for this task, its application needs to be carefully programmed according to the data and hardware characteristics.

### 3    Results

Both solubility formulations were tested simulating T$_w$ from 4ºC to 30ºC at 1ºC intervals and S from 0 ppt to 36 ppt at 1 ppt intervals. The metric k$_{H,Joh10}$/k$_{H,Sar13}$ showed better how much their estimates could diverge (Fig. 1). Afterwards, both formulations were applied to the data from the European coastal ocean. Their estimates were compared applying



the previous metric averaged over the 66h time interval using the geometric mean (Fig. 1). From the 24[th] to the 26[th]
May the water temperature at the ocean surface changed significantly and there were large fresh water inputs from the
Black Sea and the Baltic Sea (Video 1). The widest divergences were up to 4.5% in the $CO_2$ solubility estimates
associated to cooler waters, 5.8% in the $CH_4$ solubility estimates associated to both temperature extremes, and 2.1% in
the $N_2O$ solubility estimates associated to cooler and less saline waters (Fig. 1). These mismatches lead to large
differences in the estimates of greenhouse gases dissolved in the first meter below the ocean surface (Fig. 2). These
differences were estimated from $\Delta ton \cdot m^{-1} \cdot 121\ km^{-2} = 11^2 \cdot \Delta s \cdot p_{gas} \cdot P \cdot 101325 \cdot M_a/(10^9 \cdot R \cdot T)$, where $\Delta s$ was the difference in
the solubility estimated by either algorithm in its $C_w/C_a$ form at each 11 km wide cells and averaged over the 66 h time
interval. $M_a$=28.97 was the air molecular mass and $p_{gas}$ the atmospheric partial pressure of $CO_2$, $CH_4$ or $N_2O$, 390 ppm,
1.75 ppm and 0.325 ppm respectively (EPA, 2015), assuming that they were approximately uniform all over the
atmospheric SBL. These differences summed to $3.86 \times 10^6$ ton of $CO_2$, 880.7 ton of $CH_4$ and 401 ton of $N_2O$. Because
the bias of $N_2O$ changed from positive to negative with location, the overall bias was 163 ton.

The $k_w$ estimated from the E-C measurements presented a systematic bias. To detect its source, the difference
($\Delta k_w$) between the $k_w$ estimated from the E-C measurements and the one estimated from the Wan92 formulation was
compared to the potential sources of bias. Besides well correlated with $u_{10}$ (r=0.55), the $\Delta k_w$ was also well correlated
with the relative humidity (r=-0.7) and with the first (r=0.49), second (r=0.47) and third (r=0.67) terms of the WPL
correction. The distortion of the E-C flux estimates by cross-sensitivity to humidity is a common problem with open-
path IRGA, raising substantially their detection limit. The observed differences between the concentrations of $CO_2$ in
the air and in the water during our survey varied within 120 and 270$\Delta$ppm, well below the limit for a 25% error in the
flux estimates as reported by Blomquist et al. (2014) for our IRGA model, the LI-COR LI-7500. We hypothesize
whether the E-C data lacked quality to calibrate and validate the formulations. However, our formulations were close
matches to the estimates by widely used transfer velocity formulations subject to thorough calibration and validation,
which proved them reasonable estimators of the central tendency (Fig. 3). Hence, we were confident about the potential
of our newly proposed formulations to replicate the central tendency similarly well while improving the accuracy of the
estimates for each particular location.

     During this Baltic Sea sampling at the Östergarnsholm site, the SBL was generally stable (0<$Ri_b$<0.5 with a
few exceptions) and the sea-surface was little to moderately rough ($z_0$<0.49 mm). These conditions were used as
reference to estimate the elasticity of $k_w$ to its forcing functions (Fig. 4). The variables related with the SBL stability,
namely the $u_{10}$, temperature, pressure and humidity, were the variables able to induce larger changes in $k_w$. Several
renowned $u_{10}$-based formulations for the estimation of $k_w$ were used and compared with the most comprehensive
alternatives provided in our software and framework (Fig. 3). Although their estimates were close matches, there were a
few fundamental differences: the comprehensive algorithms split the data points into two distinct scatter lines, the upper
line for $k_w$ obtained under rougher sea-surfaces and the lower line for $k_w$ obtained under smother ones. The red markers
representing the ZRb03 iWLP give the best example. The $u_{10}$-based formulations were unable to perform this
adjustment to the local wave state. Their small $k_w$ fluctuations were a sole consequence of changes in water viscosity (as
estimated by the $Sc_w$) driven by changes in water temperature. These results highlight the potential of the SBL stability
and the sea-surface agitation as additional $k_w$ mediators. It is curious that the wave variables were the responsible for the
big differences between $k_w$ estimates (as shown in Fig. 3) although these were the variables to which the $k_w$ was least
elastic (as shown in Fig. 4). It demonstrates that more important than model sensitivity (or elasticity) is how much the





respective variables effectively change in the real world. There is yet the interesting detail of how the WLLP and the iWLP diverged under smoother sea-surfaces, supporting the solution suggested in the COARE 3.0 (Fairall et al., 2003) for the iterative estimation of $u_*$ and $z_0$.

305         Complementary to the analysis above, we also used the simulations of the European costal oceans to compare between the ESM standard (the Wan92) and one of our comprehensive alternatives (the iWLP-ZRb03), chosen on the basis of two factors: it was both the most elastic formulation and the one providing the closest estimates to the Wan92 (recall Fig. 3). Since the Wan92 often represented the central tendency of the iWLP-ZRb03, this choice provided the best probability that the differences between the $k_w$ estimates were due to the enhanced representation of the

environmental processes involved and not to systematic biases associated to uncertainty in the parameter estimation. Both $k_w$ estimates diverged under two particular situations (Fig. 5): (i) under low winds and unstable SBL, and (ii) under high winds and rougher sea-surfaces.

        Strong winds occurred along the European shores from the 24[th] to the 26[th] of May of 2014. Besides, the air was unusually cold for the season and colder than the sea-surface (Video 1). The upward advection of the warmer air, heated

by the sea-surface, generated turbulent eddies that enhanced mixing within the SBL. These unstable conditions were identified by $Ri_b < 0$, L tending to $^-0$ and $\psi_m < 0$ (Video 2). The mixing of the SBL enhanced $u_*$ and $k_w$ everywhere the wind blew lighter. This situation occurred more frequently and intensively nearby land masses and often associated to cooler continental breezes blowing off-shore. Its correct simulation required the estimation of the $Ri_b$, L and $\psi_m$ from the algorithms by Grachev and Fairall (1997) and Stull (1988) that account for humidity considering saturation at 0 m

heights. The $Ri_b$ estimates neglecting humidity (Lee, 1997) often yielded neutral conditions (i.e, with $Ri_b \approx 0$) or unreasonably stable SBL (i.e, with $Ri_b > 0$).

        The sea-surface agitation was very heterogenic, particularly at the coastal ocean where it attained both the highest and the lowest estimated roughness lengths (the $z_0$ in Video 3). There, the steeper waves as a consequence of shorter fetches, should extract more momentum from the atmosphere under similar $u_{10}$ conditions (Taylor and Yelland,

2001; Fairall et al, 2003). Thus, the rougher coastal ocean surfaces were expected to possess more turbulent layers through which gases were transferred at higher rates. The comprehensive formulations simulated this by increasing $u_*$ (and consequently $k_{wind}$) with $z_0$ under similar $u_z$ i.e, similar winds generate more drag when blowing over harsher sea-surfaces. Aside the rougher weather, whenever lighter wind blew over smoother sea-surfaces, the iWLP estimated much higher $z_0$ than the WLLP (video 4), demonstrating that the smooth flow was a fundamental driver for the $z_0$ under

calmer weather. This increase in $z_0$ lead to significantly higher $u_*$, often 1.5 times higher and sometimes more, anticipating a significant impact on the $k_{wind}$ estimates.

        The comprehensive formulation (i.e, ZRb03 iWLP) often estimated $k_w$ largely higher than the one estimated by the ESM standard formulation (i.e, Wan92), although it occasionally estimated lower $k_w$ (Video 5). Its largest estimates of $k_w$ were associated to unreasonably high estimates of $z_0$ that biased the subsequent results. These biased estimates of

$z_0$ could either be due to a poor calibration of the Taylor and Yelland (2001) model estimating $z_0$ from the wave field or due to biased wave field provided by the WW3-NEMO. To avoid this bias, $k_w$ was imposed a 70 cm·h[-1] ceiling, corresponding to the maximum reported in the bulk literature associated to similar wind speeds. With this restriction, the difference in the $CO_2$ volume transferred by either formulation across the $\approx 5,054,896$ km[2] of ocean surface during the 66 h was of 12997 km[3], corresponding to 33.7% of the 38551 km[3] of $CO_2$ total volume transferred using the ESM

standard formulation (Fig. 6). These differences were higher at the coastal ocean, a consequence of the factors that were



not taken into consideration by the ESM standard. The total volumes of $CH_4$ and of $N_2O$ transferred were 41156 $Km^3$ and 41158 $Km^3$, respectively. The differences were negligible between using the single layer or the double layer scheme to estimate $k_w$, even for a rather insoluble gas as is $CH_4$ (Fig. 6 and Video 5). Nevertheless, it is worth noting that it was again in the fetch limited coastal ocean where most of the bigger differences were found.


### 4     Discussion

The accurate estimation of the balances of greenhouse gases and aerosols in the atmosphere and in the oceans, as well as their fluxes across the surfaces of the coastal oceans, is an important issue for biogeosciences and Earth-System modelling (ESM). Previous estimates of $CO_2$ uptake by the global oceans done by coarse resolution implementations

diverged in about 70 % depending on the transfer velocity formulations being used (Takahashi et al., 2002), whereas the wide uncertainty in the ocean $N_2O$ source to the atmosphere mostly originated from the uncertainty in the air–water transfer velocities (Nevison et al., 1995). However, the knowledge on this subject is still limited, with plenty of room for improvement. As an example, the simpler formulations for the estimation of $k_w$ assume either a quadratic or cubic dependency from $u_{10}$ depending mostly on the sensing method, time scale and fetch at the particular location.

Furthermore, the simulation of atmosphere-ocean interactions by regional and Earth-system models, by still using these simpler formulations, are decades behind the state-of-the-art. Our work proposes a framework to incorporate this state-of-the-art in an atmosphere-ocean coupler and demonstrates that this is fundamental for reliable simulations of coastal ocean systems.

         Remarkably, both solubility formulations generally matched their estimates despite their distinct backgrounds.

Nevertheless, they did diverge in as much as 0.045 mol·$mol^{-1}$ of $CO_2$, 0.0015 mol·$mol^{-1}$ of $CH_4$ and 0.012 mol·$mol^{-1}$ of $N_2O$ (i.e, mol of gas in the ocean surface per mol of gas in the atmosphere) in some of the most sensitive situations for Earth-System modelling and satellite data processing: (i) the cooler marine waters occur closer to the poles, where the solubility pump traps greenhouse gases and carries them to the deep ocean (Sarmiento and Gruber, 2013), and (ii) the warmer and the less saline waters occurring at the coastal ocean and seas, which have regularly been observed having

greenhouse gases and aerosols dissolved in concentrations highly unbalanced with those of the atmosphere (Nevison et al., 2004; Borges et al, 2005; Barnes and Upstill-Goddard, 2011; Sarmiento and Gruber, 2013; Dutta et al., 2015; Gypens and Borges, 2015; Harley et al., 2015). Therefore, the biases in the estimated total amount of greenhouse gases in the first meter depth of the European coastal ocean during late May 2014 may be an indicator of higher global biases.

         This work showed that the accurate estimation of the transfer velocity of greenhouse gases and aerosols across

the coastal oceans' surface requires taking into consideration at least the atmospheric stability of the SBL and the sea-surface roughness, as recently suggested by Jackson et al. (2012) and Shuiqing and Dongliang (2016). Our results show that, by neglecting these factors, the simpler $u_{10}$-based formulations tend to provide lower estimates of the transfer velocity than the provided by comprehensive formulations. Similar conclusions were achieved by Jackson et al. (2012). However, the more comprehensive formulations still need improvement and validation. It is imperative to calibrate and

validate the estimation of transfer velocity from friction velocity and wind-wave breaking, and the roughness length from the wave field. All the available formulations for these specific purposes lack robust parameter estimations. Generally, there seems to be a great dependency of the available algorithms from the particular data sets that were used to calibrate them. Nevertheless, there is a general consensus that the $k_{bubble}$ term is fundamental under high wind speeds, with its estimate being central to current $k_w$ research. The latest developments have been on the dependency of $k_{bubble}$



from the interactions among the wind, the wave state, the bubble plume and the properties of the gas being transferred
(Woolf et al. 2007; Callaghan et al., 2008, 2014; Goddijn-Murphy et al., 2011, Crosswell, 2015). The effect of sea-spray
is the new buzz on this topic and only recently started emerging algorithms like the ones by Zhao et al. (2006) and Wu
et al. (2015). So far, these focused on the momentum transfer from wind to the ocean surface and the attenuation of the
friction velocity. It should be interesting to understand how the intrusion of the sea-spray on the atmosphere affects the

transfer velocity of gases, being anticipated a process symmetrical to that of the intrusion of bubbles on the ocean. The
new algorithms for the effects of surfactants are particularly concerned with the variability of the coastal ocean (Pereira
et al., 2016). These no longer associate the surfactants to the Schmidt number's exponent but rather to a coefficient
setting a proportional decay of $k_w$. The effect of sea-ice must take into consideration its distortion of the ocean surface
and its effect upon the SBL stability (Loose et al., 2014). Our coupling solution still needs to integrate the effects of the

sea-surface cool-skin and warm-layer, surfactants, rain, sea-spray and sea-ice. From these, the cool-skin and warm-layer
algorithms are the only with robust calibrations and validations, mostly done under the COARE (Fairall et al., 1996;
Fairal et al., 2003; Zeng and Beljars, 2005; Brunke et al., 2008). The addition of complexity to any coupling solution
must be carefully thought as these cannot become intricate to the point of calculus becoming unbearable for ESM
application. In particular, any algorithm demanding for-loops is unviable as it disables calculus vectorization and its

coordination with parallel processing. In our software, vectorization enabled improving calculus roughly 12× faster in a
single core.

Besides finding the appropriate algorithms and parameter values to be used by the coupler, there is also the
issue of accurately retrieving the variables mediating the gas transfer. The results showed that the $k_w$ was most elastic to
the variables related with the SBL stability, namely the $u_{10}$, temperature, pressure and humidity. Although these are

provided by the oceanic and atmospheric model components at courser vertical resolutions, they need to be transposed
to finer vertical resolutions taking into consideration the processes occurring at the sea-surface. While the $u_{10}$ is given
by the atmospheric model, the water temperature needs to account for the cool-skin and warm-layer effects and the heat
and humidity at the SBL need to account for their vertical fluxes over the sea-surface. The COARE algorithm is the
state-of-the-art for these tasks. During most of its development it focused on E-C methods to estimate the fluxes of heat

and humidity across the SBL using a framework with an intricate mathematical structure going deeper into the
simulation of the geophysical process. Given its complexity, it must be quite a challenge to perform the calculus
vectorization and parallelization required for the substantial improvement of computational speed and its application to
ESM. Only in its latter developments did the COARE explicitly addressed the fluxes of gases and the importance of
sea-surface roughness (Fairall et al, 2003; Jeffrey et al., 2010; Blomquist et al., 2006, 2014).


## 5    Code and Data Availability

Software and data related to this article provided as supplementary material. Software, data and videos related to this
article available at http://www.maretec.org/en/models/fugas

## 6    Acknowledgments:

To windguru.cz for the support providing the wave data.Work funded by ERDF Funds of the Competitiveness Factors
Operational Programme - COMPETE and by national funds from the FCT - Foundation for Science and Technology
project UID/EEA/50009/2013.





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



**8        Figures**

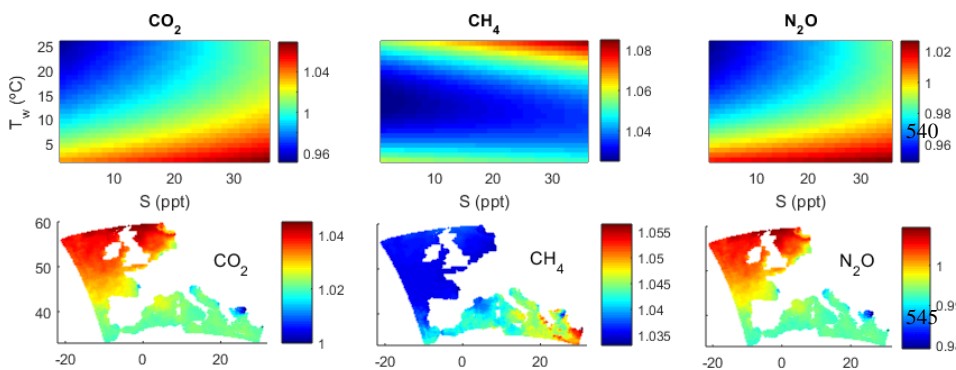

**Figure 1:Comparing solubility formulations**

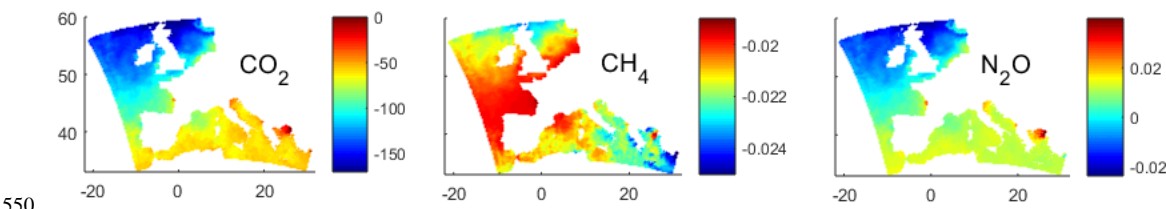


**Figure 2: Bias in the gas mass balance for the European coastal ocean**

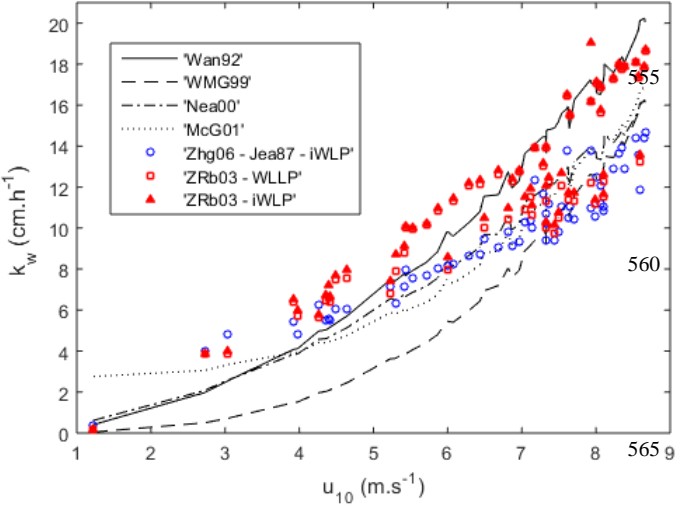

**Figure 3**: **Comparing transfer velocity**

**algorithms using observed data**





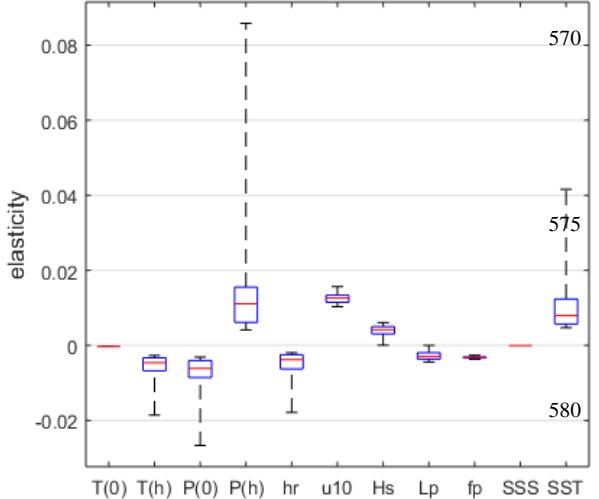



**Figure 4**: **Elasticities of the transfer velocity to its forcing functions**.



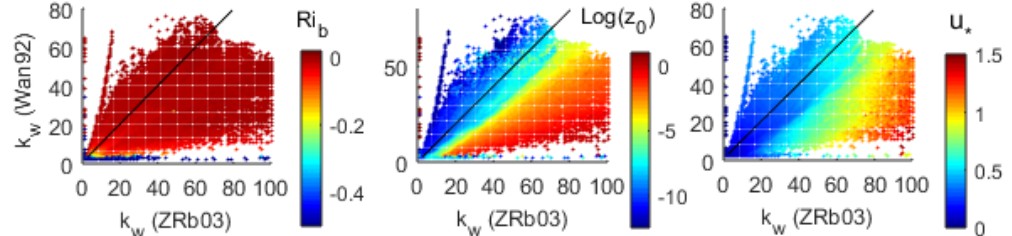

**Figure 5: Applying the modelled data about the European coastal ocean for a direct comparison between the $k_w$ estimates**.


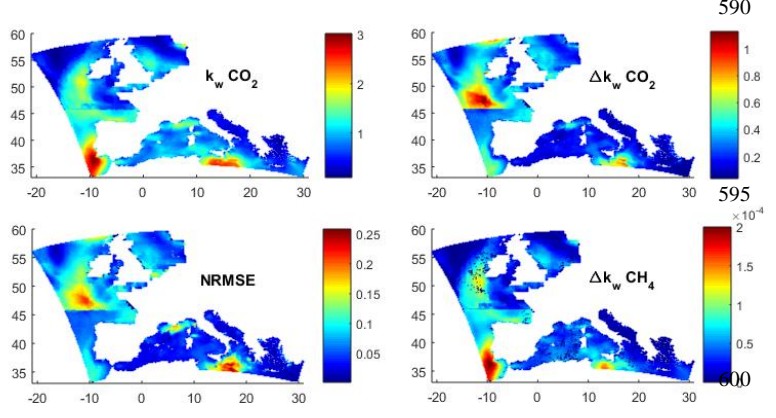


**Figure 6: Comparing transfer velocity algorithms using modelled data**




## 9      Figure legends

**Figure 1: Comparing solubility formulations**: ($k_H$) Henry's constant estimated from (Joh10) Johnson, 2010, or
(Sar13) Sarmiento and Gruber, 2013. Colorscale: $k_{H\ 'Joh10'}/k_{H\ 'Sar13'}$.

**Figure 2: Bias in the gas mass balance for the European coastal ocean**: comparing algorithm by Johnson (2010) to
compilation by Sarmiento and Gruber (2013). Colorscale: $\Delta ton \cdot m^{-1} \cdot 121\ km^{-2}$ i.e, bias in the gas mass estimated by each
algorithm ($\Delta ton$) for the first meter depth ($m^{-1}$) in 11 km wide cells ($121\ km^{-2}$).


**Figure 3: Comparing transfer velocity algorithms**. The $k_w$ estimated by renowned $u_{10}$-based formulations and by
some of the most comprehensive alternatives provided in the FuGas 2.1, using the data observed at the Baltic. 'Wan92'
- Wannninkhof (1992); 'WMG99' – Wanninkhof and McGillis (1999); 'Nea00' – Nigthingale et al. (2000); 'McG01' –
McGillis et al. (2001); 'Jea87' – Jähne et al (1987); 'Zhg06' - Zhang et al. (2006); 'ZRb03' -  Zhao et al (2003);;
assembled using the 'WLLP' – wind log-linear profile or the 'iWLP' – iteratively estimated wind log-linear profile.

**Figure 4: Elasticities of the transfer velocity to its forcing functions** $(\partial k_w/k_w)/(\partial x/x)$. The $k_w$ estimated by the
iterative wind log-linear profile (iWLP) with the Zhao et al (2003) $k_{bubble}$ term (ZRb03).

**Figure 5: Applying the modelled data about the European coastal ocean for a direct comparison between the kw
estimates** by the ESM standard and our best performing comprehensive.

**Figure 6: Comparing transfer velocity algorithms using modelled data**: ($k_w\ CO_2$) estimated by the 'iWLP' -
iterative Wind Log-Linear Profile and  the 'ZRb03' - Zhao et al. (2003) formulation, ($\Delta k_w\ CO_2$) difference between the
iWLP with ZRb03 and the Wan92' - Wanninkhof (1992) formulation, (NRMSE) Normalize Root Mean Square Error
between the iWLP with ZRb03 and the Wan92, , ($\Delta k_w\ CH_4$) difference between the single and double layer schemes
using the  iWLP with ZRb03. Colour scale: volume (or $\Delta$ volume) transferred in units of $Km^3/66h$, except for NRMSE.