# Peer review of "The FuGas 2.1 framework for atmosphere-ocean coupling in geoscientific models: improving estimates of the solubilities and fluxes of greenhouse gases and aerosols"

_Geoscientific Model Development, 2016_

## Referee Comment (RC1) · Anonymous Referee #1 · 20 Dec 2016

The manuscript "The FuGas 2.1 framework..." by Vasco M. N. C. S. Vieira and colleagues is an interesting study presenting a model which aim is to improve parameterization of air-sea gas exchange. The model is based on previous work by the same first author but on top of that, it presents the differences in gas fluxes in European coastal seas between the model and previous parameterizations.

The manuscript looks promising but still has at least one large problem (and some minor ones) which needs to be addressed before it is published. This is the very reason I suggest it needs a major revision. Namely it does not convincingly state why

the new parameterization is supposed to fit better experimental data. Lines 295-6 contain the following statement: "The red markers [in Fig. 3] representing the ZRb03 iWLP give the best example." I believe the "example" is actually "fit" but I do not see why the red triangles (representing ZRb03 iWLP) are supposed to be best fit. That is unless the authors imply that the obsolete Wannninkhof 1992 is the one they are fitting. That would be wrong because even its author suggests using his newest formula from Wanninkhof, R. (2009) [1]. This new function is closer to Nightingale et a;. 2000, also shown in Fig 3. And "Rb03 WLLP" seems closer to it. Maybe I have misunderstood the authors' intentions (for example I have no idea what the following is supposed to mean: "the comprehensive algorithms split the data points into two distinct scatter lines, the upper line for kw obtained under rougher sea-surfaces and the lower line for kw obtained under smother ones") but I have not been convinced why "ZRb03 iWLP" is supposed to be better. This certainly needs some additional arguments in any revised version.

The "renowned functions" in Fig. 3, as the authors call them, miss some other important ones like Ho et al. 2006 [2] and Sweeney et al. 2009 [3]. In fact the recommendations of a special discussion session "Relationship between wind speed and gas exchange over the ocean: Which parameterisation should I use?" on the latest SOLAS Conference (Kiel, 2015) are:

"For gas transfer of CO2 over the oceans the relationships proposed in Nightingale et al. (2000), Sweeney et al. (2007), Ho et al. (2006), and Wanninkhof et al. (2009) are recommended. They are very similar and fall within the overall uncertainty of DT measurements."

The manuscript has also some minor problems, easy to address:

- equations (13) based on Zhang et al. 2006 (eq. 3) and (18) based on Jeffrey et al. 2010 (eq. 5 and 6) are both badly mangled, most probably by the Copernicus editing software (I had the same problem with a manuscript I was a co-author of) with all "1/2"

values changed to "1.2" and other errors

- Ks is dependent on theta (equation 7) which should be explicitly shown

- the Wind Log-Linear Profile (equation 14) is actually the Monin-Obukhov similarity theory which could be directly named and cited (either the original 1954 Russian language paper or its English translation http://mcnaughty.com/keith/papers/Monin_and_Obukhov_1954.pdf)

- "the alternative model" of Liss and Slater (1974) [line 81] is not really alternative but rather needed for gases which airside resistance (1/ka) is not negligible)

- The "E-C" acronym, introduced in line 275, is never explained and has to be guessed as (non-obvious) eddy covariance / correlation

- "calculus" is used multiple times in the manuscript in the meaning of "calculation" (the Enlish language meaning is narrow and covers only derivatives and integrals)

- ocean deep waters are not formed in "pole regions" [line 46] but rather sub-polar ones, mainly Nordic Seas (Norway and Labrador Seas) of the North Atlantic

- it is not clear from the caption of Fig. 1 what is subtracted from what.

Suggested literature:

[1] Wanninkhof, R. (2014), Relationship between wind speed and gas exchange over the ocean revisited, Limnol and Oceanogr: Methods, 12, 351-362, doi:10.4319/lom.2014.12.351.

[2] Ho, D. T., C. S. Law, M. J. Smith, P. Schlosser, M. Harvey, and P. Hill (2006), Measurements of air-sea gas exchange at high wind speeds in the Southern Ocean: Implications for global parameterizations Geophys. Res. Let., 33, L16611, doi:16610.11029/12006GL026817.

[3] Sweeney, C., E. Gloor, A. R. Jacobson, R. M. Key, G. McKinley, J. L. Sarmiento,

and R. Wanninkhof (2007), Constraining global air-sea gas exchange for CO2 with recent bomb C-14 measurements, Global Biogeochem. Cycles, 21(2), GB2015 doi:10.1029/2006GB002784.

---

## Referee Comment (RC2) · Anonymous Referee #2 · 21 Dec 2016

The manuscript "gmd-2016-273" should present "an improved estimates for the solubilities and fluxes of greenhouse gases and aerosols" (see the title). However, the manuscript does not present any novel way of estimating air-sea fluxes, neither improves any of them, but rather tries to summarize what is available in the literature and to present an algorithm where many different approximation can be used. In general I am largely in favor of such manuscripts, as these can really show the state-of-the-art and our comprehension of the process described.

Nevertheless, I must admit that I found the manuscript very approximative, inconsistent

and full of mistakes (hopefully only typos). Some equations are wrong and, although this could be due to conversion of the text to pdf format, it show the lack of attention of the authors in checking the quality of the manuscript. Further, numerous acronyms are used without any explanation. As the authors seem to consistently ignoring acronym explanations, maybe the easiest solution is to add a table listing all of them at the end of the manuscript. Further, a very deep language editing is necessary before any publication.

Importantly, I would suggest to reformulate the title of the manuscript as this does not correspond to the real work presented in the manuscript. Aerosols are not included in the text, and I am puzzled to understand how a calculation of gas solubilities and piston velocities can be used to estimate fluxes of aerosols. As mentioned before, the manuscript does not present any new parametrization, but rather uses what present in the literature, showing the differences in estimating solubility and piston velocity between different formulations. Maybe a more stringent and precise title could help the reader.

Despite that, I think that the science contained in the second section (i.e. Section 3) is still interesting and valid, and would be nice to see this analysis in a well written manuscript.

As it is difficult for me to see what could be improved to make the manuscript acceptable, I will list here below some of the issue I have been finding in the manuscript.

**title** FuGas2.1 is mentioned in the title, but the acronym is NEVER explained in the entire manuscript.

**line 39** Not all Regional models have land, ocean, atmosphere and cryosphere components

**Introduction** Probably few more citation would help the reader.

**line 51** In all the text there is a consistent usage of acronyms that were never explained before. Example: IPCC, MPI, CMCC.

**line 55** Here's a good example of acronym explanation missing: What is MOHID? Why is that important in this text/section? I really appreciate that MOHID allows to use different formulations, but, is this really important?

**line 58** As you mentioned that "there are many other simpler formulation" it would be probably good to list some of them.

**line 60** "...adjustment to their specific data": which data? What do you mean with such sentence?

**line 81** Well, also the 3-layer model is present (see Cen-Lin and Tzung-May. (2013)). Which model are you using in all the flux calculations afterwards?

**line 84** Actually the transfer velocity is NOT averaged over both layer. The formulation follows the Fick's law of diffusion, i.e. assuming that the transport across the thin layers is in a steady state.

**section 2.1** Maybe it would be good to make two subsection for the two solubility formulation, so that the reader immediately understand which parametrization will be compared afterward.

**line 106** What do you mean with "alternative chemistry background"?

**line 116** There are typos in the equation (6). Some $1$ are present making 298.15 equal to 1298.15

**line 144** As you mention wind and bubble (white caps), what about precipitation (i.e. rain)? See Ho et al. (1997).

**line 164** Here the Schmidt number is to the power of 12. Should be $1/2$.

**line 181** As in many equations in the manuscript, here as well all the terms of the equation are not fully explained. The meaning of $L_p$ comes only on page 7, line 236.

**line 186** What do you mean with "in its turn" ?

**line 188** . I appreciated that you are now listing terms of equation (17). However, you also list terms which do not exist in the equation, such as $T_z$, $P_z$ and $q_z$. This is very hard for the reader, as most of the equations are not well explained and other have additional explanations which should not be present...

**line 203** "..we compared between..." . Where are the results presented?

**line 207** Again here you have typos with the power. I expect these to be $1/2$ and not 12.

**line 224** I do not think that this title is appropriate. You do not present any coupler, but rather you are describing the simulated data you will be using for your algorithm.

**line 261** Nice that you explain the metric. However these variable (i.e. $K_{HJohn10}$ and $K_{HSar13}$ are nowhere explained before. The only explanation is in the figure labels. Although one could guess where they come from, this should be better explained.

**line 269** I did not understand where this equation comes from. Do you need a piston velocity to calculate the differences? If so, how do you calculate that? What about the concentration in the water? Do you assume that equal to zero? Could you please formulate better this calculation?

**line 275** "E-C" has never described before. I could guess it refers to E(ddy)-C(ovariance) method but it is impossible to know for sure.

[Figure]

**line 296** As before, the "ZRb03 iWLP" formulation is based on a mysterious parametrization, that the reader can only guess from the sequence of symbol and letter. Probably you should list them and explain exactly on what they are based. A table could also help.

**line 355** I disagree that ESM use simple approach. Please see Pozzer et al. (2006) and the model AIRSEA.

**line 359** This line does not make any sense to me: what do you mean with "both formulations matched their estimates"???

**line 360** Would be nice to put this number in contest of numerical error. Does this difference in solubility really play a role?

**line 368** This can be easily tested, using he Takahashi et al. (2009) compilation and calculating the effect for different formulation (for $CO_2$). However, here the discussion must be taken cautiously: in fact, due to their coarse resolution, Earth System Models do not represent coastal area very well. Is that important at all in the overall, for example, $CO_2$ budget? How much is "coastal area" compared to open ocean. Can this difference really influence the calculation of carbon cycle in global model?

**line 394** I do not think you can make such general statement... "do loops" exists also in vectorised and/or parallel processed algorithm.

**line 499** Maybe the reference is wrong as I though that the book of Sarmiento was published in 2006 and not 2013. Please check.

**Figure 4** It is not explained what the bars represent. How was the "elasticity" range calculated? Maybe additional explanation in the text may help the reader.

---

## Author Comment (AC1) · 9 Jan 2017

Referee Comment: The manuscript "The FuGas 2.1 framework..." by Vasco M. N. C. S. Vieira and colleagues is an interesting study presenting a model which aim is to improve parameterization of air-sea gas exchange. The model is based on previous work by the same first author but on top of that, it presents the differences in gas fluxes in European coastal seas between the model and previous parameterizations. The manuscript looks promising but still has at least one large problem (and some minor ones) which needs to be addressed before it is published. This is the very reason I

suggest it needs a major revision. Authors' reply: We kindly acknowledge the revision with constructive comments that were a significant help improving the manuscript.

Referee Comment: Namely it does not convincingly state why the new parameterization is supposed to fit better experimental data. Lines 295-6 contain the following statement: "The red markers [in Fig. 3] representing the ZRb03 iWLP give the best example." I believe the "example" is actually "fit" but I do not see why the red triangles (representing ZRb03 iWLP) are supposed to be best fit. That is unless the authors imply that the obsolete Wannninkhof 1992 is the one they are fitting. That would be wrong because even its author suggests using his newest formula from Wanninkhof, R. (2009) [1]. This new function is closer to Nightingale et a;. 2000, also shown in Fig 3. And "Rb03 WLLP" seems closer to it. Maybe I have misunderstood the authors' intentions (for example I have no idea what the following is supposed to mean: "the comprehensive algorithms split the data points into two distinct scatter lines, the upper line for kw obtained under rougher sea-surfaces and the lower line for kw obtained under smother ones") but I have not been convinced why "ZRb03 iWLP" is supposed to be better. This certainly needs some additional arguments in any revised version.

Authors' reply: The reviewer is quite right. Besides fair, his demand was quite fortunate as it drove us into significant improvements on the manuscript, and particularly in this section. From these improvements resulted a much stronger manuscript, more interesting to the scientific community, and better able to answer questions raised by reviewer #2. Our kind regards to reviewer #1. The "example" was not to be interpreted as "fit".

Referee Comment: The "renowned functions" in Fig. 3, as the authors call them, miss some other important ones like Ho et al. 2006 [2] and Sweeney et al. 2009 [3]. In fact the recommendations of a special discussion session "Relationship between wind speed and gas exchange over the ocean: Which parameterisation should I use?" on the latest SOLAS Conference (Kiel, 2015) are: "For gas transfer of CO2 over the oceans the relationships proposed in Nightingale et al. (2000), Sweeney et al. (2007),

Ho et al. (2006), and Wanninkhof et al. (2009) are recommended. They are very similar and fall within the overall uncertainty of DT measurements."

Authors' reply: Done. The formulations by Nightingale et al. (2000) and Sweeney et al. (2007) were already included in the software. The formulations by Ho et al. (2006) and Wanninkhof et al. (2009) were presently included. They are all in the new Fig.3.

Referee Comment: The manuscript has also some minor problems, easy to address: - equations (13) based on Zhang et al. 2006 (eq. 3) and (18) based on Jeffrey et al. 2010 (eq. 5 and 6) are both badly mangled, most probably by the Copernicus editing software (I had the same problem with a manuscript I was a co-author of) with all "1/2" values changed to "1.2" and other errors.

Authors' reply: Done. Actually, we realize this problem occurred prior to submission due to change of Windows based Equation Editors (Microsoft Office 2007 is terrible!). Luckily, the reviewer detected it. We found more similar typos and corrected them all.

Referee Comment: - Ks is dependent on theta (equation 7) which should be explicitly shown

Authors' reply: But it was (line 113 of old manuscript).

Referee Comment: - the Wind Log-Linear Profile (equation 14) is actually the MoninObukhov similarity theory which could be directly named and cited (either the original 1954 Russian language paper or its English translation http://mcnaughty.com/keith/papers/Monin_and_Obukhov_1954.pdf)

Authors' reply: Done. We believe Stull (1988) is another famous and adequate citation. We included both. Without any disregard for Monin and Obukhov, we had not done it before as the common policy by publications is no need for citations on well-known, unanimously accepted theories, laws, methods, etc. And how do we cite it? Is it OK how we did?

Referee Comment: - "the alternative model" of Liss and Slater (1974) [line 81] is not

really alternative but rather needed for gases which airside resistance (1/ka) is not negligible)

Authors' reply: Done. Changed accordingly.

Referee Comment: - The "E-C" acronym, introduced in line 275, is never explained and has to be guessed as (non-obvious) eddy covariance / correlation

Authors' reply: Done. Changed accordingly.

Referee Comment: - "calculus" is used multiple times in the manuscript in the meaning of "calculation" (the Enlish language meaning is narrow and covers only derivatives and integrals)

Authors' reply: Done. Changed accordingly.

Referee Comment: - ocean deep waters are not formed in "pole regions" [line 46] but rather sub-polar ones, mainly Nordic Seas (Norway and Labrador Seas) of the North Atlantic.

Authors' reply: Done. Changed accordingly.

Referee Comment: - it is not clear from the caption of Fig. 1 what is subtracted from what.

Authors' reply: Nothing is being subtracted. The colour scale is a quotient comparing the kH predicted by either formulations. Done. Changed accordingly.

---

## Author Comment (AC2) · 9 Jan 2017

Referee Comment: The manuscript "gmd-2016-273" should present "an improved estimates for the solubilities and fluxes of greenhouse gases and aerosols" (see the title). However, the manuscript does not present any novel way of estimating air-sea fluxes, neither improves any of them, but rather tries to summarize what is available in the literature and to present an algorithm where many different approximation can be used. Authors' reply:

1. The objective of the FuGas framework is allowing the researchers to test and compare alternative algorithms for the estimation of the solubilities and transfer velocities, which together set the atmosphere-ocean fluxes of greenhouse gases and DMS, with an impact on global warming.

2. The FuGas is novel when it "tries to summarize what is available in the literature and to present an algorithm where many different approximation can be used".

3. The closest to the FuGas is the EngineFlux, which is not even close to the amount of possibilities provided by the FuGas for transfer velocities, is not able to fill in the gaps by using the formulations possible given the data constrains at particular locations, and does not consider alternative formulations for solubilities. The FuGas is novel in all these aspects.

4. The code relative to Equation 10 considers simultaneously kwind, kbubble and kcurrent, which is novel. The reviewer #2 will not find it anywhere else. The kcurrent was not explicitly included in the manuscript due to previous negative feedback about presenting a term that was not being tested. In reply to reviewer #2, we got it back in.

5. Equations 10-16, with focus on equation 14, represent a novel way to integrate the effects of wind speed, atmospheric stability and sea-state on the estimation of kw. The reviewer #2 will not find this solution anywhere else. This was clearly recognized by reviewer #1 and in previous reviews by Dr Wanninkhof and Dr Johnson.

6. The results summarized in Fig.3 were significantly improved in reply to reviewers #1 and #2. Consequently, it is demonstrated that our solution represents a major improvement.

Referee Comment: In general I am largely in favor of such manuscripts, as these can really show the state-of-the-art and our comprehension of the process described. Nevertheless, I must admit that I found the manuscript very approximative, inconsistent and full of mistakes (hopefully only typos). Some equations are wrong and, although this could be due to conversion of the text to pdf format, it show the lack of attention of

the authors in checking the quality of the manuscript.

Authors' reply: The mistakes were typos due to format conversion. The equations are all correct in the docx file that we attach. They have always been correct in the software. Nevertheless, the software has been upgraded with new formulations suggested by reviewer #1. The software upgrades (FuGas 2.2) are presented in a separate comment.

Referee Comment: Further, numerous acronyms are used without any explanation. As the authors seem to consistently ignoring acronym explanations, maybe the easiest solution is to add a table listing all of them at the end of the manuscript. Further, a very deep language editing is necessary before any publication.

Authors' reply: Done.

Referee Comment: Importantly, I would suggest to reformulate the title of the manuscript as this does not correspond to the real work presented in the manuscript.

Authors' reply: Partially done as we partially disagree with the reviewer.

Referee Comment: Aerosols are not included in the text, and I am puzzled to understand how a calculation of gas solubilities and piston velocities can be used to estimate fluxes of aerosols.

Authors' reply: We meant DMS, which is released from the ocean to the atmosphere and quickly degenerates to sulphide aerosols. These are the only aerosols with a cooling effect on the atmosphere. The FuGas allows for the estimation of the DMS fluxes applying the DMS solubility constants provided by Sander (1999, 2015). The transfer velocities and fluxes of DMS have been studied by Hubert et al. (2004,2010) Blomquist et al. (2006), Fairall et al (2006), Vlahos and Mohonan (2009), Vlahos et al. (2011), Bell (2013, 2015), among others. Done. Changed accordingly in the title and text.

Referee Comment: As mentioned before, the manuscript does not present any new

parametrization, but rather uses what present in the literature, showing the differences in estimating solubility and piston velocity between different formulations. Maybe a more stringent and precise title could help the reader.

Authors' reply: The manuscript does present a new parameterization, as replied above. Nevertheless, the title was changed as replied above.

Referee Comment: Despite that, I think that the science contained in the second section (i.e. Section 3) is still interesting and valid, and would be nice to see this analysis in a well written manuscript. As it is difficult for me to see what could be improved to make the manuscript acceptable, I will list here below some of the issue I have been finding in the manuscript.

Authors' reply: We acknowledge the help to improve the manuscript. Consequently, we changed significantly all sections. The manuscript is much sounder and reads much better. Our kind regards to reviewer #2.

Referee Comment: title FuGas2.1 is mentioned in the title, but the acronym is NEVER explained in the entire manuscript.

Authors' reply: Done. In part, because it is also an analogy with the Portuguese word spelled equally and meaning "leaks" or "escapes". However, only Portuguese speaking readers would understand.

Referee Comment: line 39 Not all Regional models have land, ocean, atmosphere and cryosphere components.

Authors' reply: The sentence identifies the model components but does not state it is mandatory to have them all.

Referee Comment: Introduction Probably few more citation would help the reader.

Authors' reply: A few more citations were added to the work

Referee Comment: line 51 In all the text there is a consistent usage of acronyms that

were never explained before. Example: IPCC, MPI, CMCC.

Authors' reply: Done

Referee Comment: line 55 Here's a good example of acronym explanation missing: What is MOHID? Why is that important in this text/section? I really appreciate that MOHID allows to use different formulations, but, is this really important?

Authors' reply: MOHID has no acronym explanation. But what it is was well explained in the same sentence. We also upgraded MOHID with the COARE algorithm and this is debated in the discussion.

Referee Comment: line 58 As you mentioned that "there are many other simpler formulation" it would be probably good to list some of them.

Authors' reply: Done

Referee Comment: line 60 "...adjustment to their specific data": which data? What do you mean with such sentence?

Authors' reply: The data used in their calibration. Explanation inserted in the text.

Referee Comment: line 81 Well, also the 3-layer model is present (see Cen-Lin and Tzung-May. (2013)). Which model are you using in all the flux calculations afterwards?

Authors' reply: We used the single layer except were explicitly mentioned the double layer. That occurred in the comparison between both models and is presented in the last paragraph of the results. It was shown in Fig.6 as well as in video 5. Differences were negligible when using greenhouse gases. We changed fig.6 to be more focused on the essential. Nevertheless, the comparison between both models is still presented in the results and shown in Video 5.

Referee Comment: line 84 Actually the transfer velocity is NOT averaged over both layer. The formulation follows the Fick's law of diffusion, i.e. assuming that the transport across the thin layers is in a steady state.

Authors' reply: Done. Changed.

Referee Comment: section 2.1 Maybe it would be good to make two subsection for the two solubility formulation, so that the reader immediately understand which parametrization will be compared afterward.

Authors' reply: Each solubility formulation is already well separated in its own paragraph. Each specific paragraph starts by identifying the formulation:

1. "Sarmiento and Gruber (2013) compiled the algorithm . . . "

2. "Johnson (2010) developed an algorithm . . ."

We think this is enough

Referee Comment: line 106 What do you mean with "alternative chemistry background"?

Authors' reply: As explained in the text, the algorithm compiled by Sarmiento and Gruber (2013) is based on the ideal gas law while the algorithm compiled by Sander (1999, 2015) and Johnson (2010) is based on the molecular and thermodynamic properties of the water, its solutes and the specified gas.

Referee Comment: line 116 There are typos in the equation (6). Some 1 are present making 298.15 equal to 1298.15

Authors' reply: Done.

Referee Comment: line 144 As you mention wind and bubble (white caps), what about precipitation (i.e. rain)? See Ho et al. (1997).

Authors' reply: Rain was mentioned in the introduction and in the second line of this section (2.2). But we cannot include it in the sentence (formerly in line 144) as we are presenting the framework, which still does not account for rain. The importance of upgrading the framework with the effect of rain in kw was already in the Discussion.

Referee Comment: line 164 Here the Schmidt number is to the power of 12. Should be 1/2.

Authors' reply: Another typo that was rectified.

Referee Comment: line 181 As in many equations in the manuscript, here as well all the terms of the equation are not fully explained. The meaning of Lp comes only on page 7, line 236.

Authors' reply: Done. Changed.

Referee Comment: line 186 What do you mean with "in its turn" ?

Authors' reply: Deleted.

Referee Comment: line 188 . I appreciated that you are now listing terms of equation (17). However, you also list terms which do not exist in the equation, such as Tz, Pz and qz. This is very hard for the reader, as most of the equations are not well explained and other have additional explanations which should not be present...

Authors' reply: Done.

Referee Comment: line 203 "..we compared between..." . Where are the results presented?

Authors' reply: The results were presented in the last paragraph of the results, in Fig.6 and in Video 5.

Referee Comment: line 207 Again here you have typos with the power. I expect these to be 1/2 and not 12.

Authors' reply: Another typo that was rectified.

Referee Comment: line 224 I do not think that this title is appropriate. You do not present any coupler, but rather you are describing the simulated data you will be using for your algorithm.

Authors' reply: Done.

Referee Comment: line 261 Nice that you explain the metric. However these variable (i.e. KHJohn10 and KH Sar13 are nowhere explained before. The only explanation is in the figure labels. Although one could guess where they come from, this should be better explained.

Authors' reply: Done.

Referee Comment: line 269 I did not understand where this equation comes from. Do you need a piston velocity to calculate the differences? If so, how do you calculate that? What about the concentration in the water? Do you assume that equal to zero? Could you please formulate better this calculation?

Authors' reply: Done. Explained in the methods, Section 2.1 (Solubility) after presenting both formulations.

Referee Comment: line 275 "E-C" has never described before. I could guess it refers to E(ddy)- C(ovariance) method but it is impossible to know for sure.

Authors' reply: Done.

Referee Comment: line 296 As before, the "ZRb03 iWLP" formulation is based on a mysterious parametrization, that the reader can only guess from the sequence of symbol and letter. Probably you should list them and explain exactly on what they are based. A table could also help.

Authors' reply: Fig.3 was changed substantially and this is no longer an issue.

Referee Comment: line 355 I disagree that ESM use simple approach. Please see Pozzer et al. (2006) and the model AIRSEA.

Authors' reply: Done. In fact, the AIRSEA is more elaborated than other couplers in ESM and regional models, and we were unfair with its developers. In the meanwhile, this whole section had already been changed in reply to reviewer #1 and this sentenced

was deleted.

Referee Comment: line 359 This line does not make any sense to me: what do you mean with "both formulations matched their estimates"???

Authors' reply: Done. It means that both solubility formulations generally estimated similar solubilities. Changed accordingly in the text.

Referee Comment: line 360 Would be nice to put this number in contest of numerical error. Does this difference in solubility really play a role?

Authors' reply: We absolutely agree. But that is a whole new study and publication; which, by the way, we intend to do.

Referee Comment: line 368 This can be easily tested, using the Takahashi et al. (2009) compilation and calculating the effect for different formulation (for CO2). However, here the discussion must be taken cautiously: in fact, due to their coarse resolution, Earth System Models do not represent coastal area very well. Is that important at all in the overall, for example, CO2 budget? How much is "coastal area" compared to open ocean. Can this difference really influence the calculation of carbon cycle in global model?

Authors' reply: 1. As above, it is a whole new work that makes part of our future plans. 2. Besides the Takahashi (2009) compilation, there are other alternatives as the SO-CAT or applying algorithms to the sea-surface data soon to be provided by Copernicus Marine for the whole world at fine resolution. 3. The importance of the coastal ocean was better debated in a previous version of this manuscript in Biogeoscience Discussions. That presentation was a bit off from the objective of this work and thus was significantly shortened. Nevertheless, (i) according to Smith and Hollibaugh (1993), although the coastal ocean is ≈5% of the global oceans, ≈50% of the CO2 exchange occurs there; (ii) A large proportion of the atmosphere-ocean exchange of CH4 and N2O occurs in the coastal ocean and represents release to the atmosphere.

Referee Comment: line 394 I do not think you can make such general statement... "do loops" exists also in vectorised and/or parallel processed algorithm.

Authors' reply: Yes, they do, but they are very conflictive. It is not hard to vectorized and parallelize a simple Monte Carlo simulation applied to a simpler numerical or statistical test using a data set that has a simple structure (irrespective of its size). But it is a whole different thing to vectorized and parallelize when you have an intricate algorithm with parent scripts and child scripts, nested loops and functions, etc. Parallelization becomes unstable and crashes with apparently no reason. As an example, it is virtually impossible to vectorized and parallelize nested for-loops. This was better explained in the text.

Referee Comment: line 499 Maybe the reference is wrong as I though that the book of Sarmiento was published in 2006 and not 2013. Please check.

Authors' reply: The book has 4 editions, including one from 2006 and another from 2013. Both are available in amazon.com. I read the one from 2013.

Referee Comment: Figure 4 It is not explained what the bars represent. How was the "elasticity" range calculated? Maybe additional explanation in the text may help the reader.

Authors' reply: Done in the figure legend.

---

## Author Comment (AC3) · 9 Jan 2017

**The FuGas 2.2 framework for atmosphere-ocean coupling in geoscientific models: **comparing and** improving **algorithms for the** estimates of the solubilities and fluxes of greenhouse gases and DMS**

[revised manuscript text omitted]

The mismatches between both algorithms lead to differences in the estimates of greenhouse gases dissolved in the first meter below the ocean surface, which were calculated from $\Delta\text{ton}\cdot m^{-1}\cdot 121\ km^{-2} = 11^2\cdot\Delta s\cdot p_{gas}\cdot P\cdot 101325\cdot M_a/(10^9\cdot R\cdot T)$. The $\Delta s$ was the difference in the solubilities estimated by both algorithms and converted to the $C_w/C_a$ form. Hence, $\Delta s = 1/k_{H\text{'Sar13'}} - 1/k_{H\text{'Joh10'}}$, and because $C_a$ was equal among them, the $\Delta s = (C_{w\text{'Joh10'}} - C_{w\text{'Sar13'}})/C_a$. This difference of $\text{mol}\cdot m^{-3}$ of gas in the water per $\text{mol}\cdot m^{-3}$ of gas in the air at each cell was averaged over the 66 h time interval. In order to convert from mols to grams in the water we multiplied by the molar mass of the specific gas ($M_a$), which is 44.01 for $CO_2$, 16.043 for $CH_4$ and 44.013 for $N_2O$. Then, we divided by $10^6$ to convert from grams to tons. We still needed to

145 estimate $C_a$ from the atmospheric pressure (P) and the partial pressure ($p_{gas}$) of $CO_2$, $CH_4$ or $N_2O$, 390 ppm, 1.75 ppm and 0.325 ppm respectively (EPA, 2015), assuming that they were approximately uniform all over the atmospheric surface boundary layer (SBL). Using the ideal gas law, we divided by R and T (in Kelvin), multiplied by 101325 to convert Pascal to atm and divided by $10^6$ to re-scale from ppm to unity. Finally, we multiplied by $(11 \text{ km})^2$ to have the total difference in mass dissolved in the first meter below the surface of 11 km wide cells.

**2.2    Transfer velocity**

[revised manuscript text omitted]

$$\frac{z_0}{H_s} = 1200 \cdot \left(\frac{H_s L_p}{L_p}\frac{H_s}{L_p}\right)^{4.5} \tag{15}$$

$$z_0 = 1200 \cdot H_s \left(\frac{H_s}{L_p}\right)^{4.5} + \frac{0.11\nu_a}{u_*} \tag{16}$$

Atmospheric stability characterized the tendency of the surface boundary layer (SBL) to be well mixed (unstable SBL with $\psi_m<0$) or stratified (stable SBL with $\psi_m>0$). The $\psi_m$ was inferred from the 'bulk Richardson number' ($Ri_b$: Eq. 17), weighting the air vertical heat gradient and kinetic energy, and required the air virtual potential temperature $(T_v)$, estimated from air temperature (T in ºC), air pressure (P in atm), humidity (dimensionless) and the gravitational acceleration constant (g). Grachev and Fairall (1997) estimated $T_v=T_p(1+0.61q)$, where $T_p$ is the air potential temperature and q the observed specific humidity  or from. Stull (1988) estimated $T_v=T_p(1+0.61·r_{sat}·h_r+r_l)$, where $r_{sat}$ is the water vapour mixing ratio at saturation, $h_r$ is the observed relative humidity and $r_l$ the observed liquid water mixing ratio . $T_p$ (in ºC) was estimated from $T_p = T_k(1000/(1013.25P))^{0.284}$, where $T_k$ is temperature (Kelvin). The $r_{sat} = 0.622e_{sat}/(101.32501P-e_{sat})$ and the $ln(e_{sat}) = ln0.61078 +17.2694T/(T_k-35.86)$. Alternatively, Lee (1997) estimated the $Ri_b$ directly from the air potential temperature neglected humidity . The wind velocity ($u_z$), temperature ($T_z$), pressure ($P_z$) and humidity ($q_z$) z meters above sea-surface were given by the WRF second level. The wind velocity at $z_0$ ($u_0$) was set to the theoretical $u_0=0$. Temperature at the height of 0 m ($T_0$) was given by the SST (Grachev and Fairall, 1997; Fairal et al., 2003; Brunke et al., 2008) without rectification for cool-skin and warm-layer effects due to the lack of some required variables. Yet, these effects tend to compensate each other (Brunke et al., 2008; Fairall et al., 1996; Zeng and Beljars, 2005). Air pressure at 0 m ($P_0$) was given by the WRF at the lower first level (at roughly 0 m). Humidity at 0 m ($q_0$) was set to the saturation level at $P_0$ and $T_0$ using $q=r_{sat}/(1+r_{sat})$ (Grachev and Fairall, 1997). The $Ri_b$ was used to estimate the length L from  the Monin-Obukhov's similarity theory, a discontinuous exponential function tending to $\pm \infty$ when $Ri_b$ tends to $\pm0$ and tending to $\pm0$ when $Ri_b$ tends to $\pm\infty$. $Ri_b$ and L were used to estimate $\psi_m$ following Stull (1988) or Lee (1997) algorithms.

$$Ri_b = \frac{g \Delta T_v T \Delta z_i}{T_v T \cdot u_z^2}$$

(17)

 Its transfer velocity of mildly soluble gases, besides taking into consideration the molecular crossing of the water-side surface layer, should also take into consideration the molecular crossing of the air-side surface layer (Liss and Slater, 1974, Wanninkhof et al., 2009; Johnson, 2010). We compared between the use of the traditional single layer model and the double layer "thin film" model (Liss and Slater, 1974), the later estimating the air-side transfer velocity ($k_a$)  from the COARE formulation as in Eq. 18 (Jeffrey et al., 2010). CD is the drag coefficient and $Sc_a$ the Schmidt number of air, which were determined for a given temperature and salinity following Johnson (2010).

[revised manuscript text omitted]

(i)  invoking the parallel processing toolbox was time consuming,

(ii)  replicating over 1Gb ram was time consuming,

(iii)  when running the calculations with other programs on the background  the 4Gb ram memory was soon exhausted. Thereafter, the use of virtual memory in the hard-disk stalled enormously the calculations. To prevent it the following actions were implemented,

(iv) Programs like the antivirus, backup tools, Office, Skype, Dropbox, etc were shut down in the Task Manager,

(v)  the spmd were split into several sequential code blocks and in-between the variables no longer necessary were deleted. This spmd fragmentation was time consuming.

In conclusion, there is no perfect solution for  parallel computing, and although spmd is the best strategy available for this task, its application needs to be carefully programmed according to the data and hardware characteristics. Optimizing the data management was one fundamental improvement between the 2.1 and the 2.2 versions of the FuGas. The simulations of the European coastal ocean no longer exhausted the ram memory and did not even use more than 70% of it (using above 90% becomes critical). Consequently, the computation time improved from ≈12 min to ≈4 min.

**3    Results**

The solubility formulations were compared simulatthe range of environmental conditions commonly found in nature: $T_w$ ranged from 4ºC to 30ºC at 1ºC intervals while S ranged from 0 ppt to 36 ppt at 1 ppt intervals. The ratio between the solubilities estimated by each formulation (i.e., $k_{H,Joh10}/k_{H,Sar13}$) showed better how much the could diverge (Fig. 1). Afterwards, both formulations were applied to the conditions observed at the European coastal ocean during the experiment. The estimated solubilities were compared from their ratio averaged over the 66 h time interval using the geometric mean (Fig. 1). From the 24$^{th}$ to the 26$^{th}$ May the water temperature at the ocean surface changed significantly and there were large fresh water inputs from the Black Sea and the Baltic Sea (Video 1). The widest divergences were up to 4.5% in the $CO_2$ solubility estimates associated to cooler waters, 5.8% in the $CH_4$ solubility estimates associated to both temperature extremes, and 2.1% in the $N_2O$ solubility estimates associated to cooler and less saline waters (Fig. 1). These mismatches lead to large differences in the estimates of greenhouse gases dissolved in the first meter below the ocean surface (Fig. 2). ~~These differences summed to 3.86×10⁶ ton of CO₂, 880.7 ton of CH₄ and 401 ton of N₂O. Because the bias of N₂O changed from positive to negative with location, the overall bias was 163 ton. These differences were estimated from Δton·m⁻¹·121 km⁻² = 11³·Δs·p_gas·P·101325·M_a/(10⁹·R·T), where Δs was the difference in the solubility estimated by either algorithm in its C_w/C_a form at each 11 km wide cells and averaged over the 66 h time interval. M_a=28.97 was the air molecular mass and p_gas the atmospheric partial pressure of CO₂, CH₄ or N₂O, 390 ppm, 1.75 ppm and 0.325 ppm respectively (EPA, 2015), assuming that they were approximately uniform all over the atmospheric SBL.~~Integrated over space, these differences summed to 1.17×10$^5$ ton of $CO_2$, 7374.5 ton of $CH_4$ and 25.1 ton of $N_2O$.

During the Baltic Sea sampling at the Östergarnsholm site the observed Δppm varied within 120 and 270, well below the limit for a 25% error in the flux estimates as reported by Blomquist et al. (2014) for our IRGA model, the LI-

COR LI-7500. Even so,

~~The $k_w$ estimated from the E-C measurements presented a systematic bias. To detect its source, the difference ($\Delta k_w$) between the $k_w$ estimated from the E-C measurements and the one estimated from the Wan92 formulation was compared to the potential sources of bias. Besides well correlated with $u_{10}$ (r=0.55), the $\Delta k_w$ was also well correlated with the relative humidity (r=-0.7) and with the first (r=0.49), second (r=0.47) and third (r=0.67) terms of the WPL correction. The distortion of the E-C flux estimates by cross-sensitivity to humidity is a common problem with open-path IRGA, raising substantially their detection limit. The observed differences between the concentrations of $CO_2$ in the air and in the water during our survey varied within 120 and 270$\Delta$ppm, well below the limit for a 25% error in the flux estimates as reported by Blomquist et al. (2014) for our IRGA model, the LI COR LI-7500. We hypothesize whether the E-C data lacked quality to calibrate and validate the formulations. However, our formulations were close matches to the estimates by widely used transfer velocity formulations subject to thorough calibration and validation, which proved them reasonable estimators of the central tendency (Fig. 3). Hence, we were confident about the potential of our newly proposed formulations to replicate the central tendency similarly well while improving the accuracy of the estimates for each particular location.~~

the $k_w$ estimated from the Eddy-Covariance (E-C) measurements  were close matches to the $k_w$ estimated by both generalistic and comprehensive algorithms (Fig.3).  The mismatches found in previous versions of this report were consequence of humidity mistakenly input in incorrect units into the E-C matlab script. The mismatches currently seen at lower wind speeds may be due to failure of the E-C measurements under atmospherically stable conditions, a problem well known to affect E-C methods. 5 with a couple of exceptions above 0.09) while the sea-surface was little to moderately rough ($z_0<0.49$ mm). These conditions were used as reference to estimate the elasticity of $k_w$ to the environmental variables (Fig. 4). The variables related with the SBL stability, namely the $u_{10}$, temperature, pressure and humidity, were able to induce larger changes in $k_w$. However, the SBL stability changed little during this experiment whereas the sea-state change considerably, with a calmer period during which the sea-surface was smoother and a harsher period during which the sea-surface was rougher (Fig.3). Hence, during this experiment the sea-state had a greater impact on the $k_w$ than the atmospheric stability. The $k_w$ dependency on the sea-state is well-known as it is thought that $k_w$ scales with the turbulent kinetic energy dissipation at the sea-surface ($\varepsilon$) and that this is better reflected by the sea-state (Soloviev et al., 2007, Wang et al., 2015). Accordingly, the COARE 3.0 included the wave state in the estimation of the roughness parameters essential for the transfer of mass, heat and momentum (Fairal et al. 2003) while Frew et al. (2004) observed a remarkable correlation between the $k_w$ and small scale waves. Our comprehensive algorithms adjusted to the sea-state splitting the $k_w$ estimates into two distinct groups relative to each period. The $k_w$ estimated for the rougher sea-states scattered along a steeper line placed above the $k_w$ estimated for the smoother sea-states. The $k_w$ estimated from the E-C measurements tended to follow this same pattern (Fig.3), and episodic departure from it may be inherent to E-C natural variability.  were used and compared with the most comprehensive alternatives provided in our software and framework (Fig. 3). Although their estimates were close matches, there were a few

 The generalistic $u_{10}$-based formulations were unable to perform this adjustment to the local wave state. Their small $k_w$ fluctuations were a sole consequence of changes in water viscosity (as estimated by the $Sc_w$) driven by changes in water temperature. Furthermore, the Wan92 formulation and our comprehensive formulations adjusted remarkably well under rougher seas, whereas the Cea96 formulation calibrated with data from the Parker river estuary and our comprehensive formulations adjusted remarkably well under lighter winds and/or smoother seas (Fig.3). These fits clearly show an ability of our comprehensive algorithms to adjust to the local conditions that cannot be met by the generalistic $u_{10}$-based formulations. This is not a minor detail: at $u_{10} \approx 8$ $m \cdot s^{-1}$ the $k_w$ estimated from wind dragging over rougher or over smoother sea-surfaces differed $\approx 31\%$ while at $u_{10} \leq 4$ $m \cdot s^{-1}$ the $u_{10}$-based formulations estimated less than 50% of the $k_w$ estimated by our comprehensive algorithms. Furthermore, under the lowest winds the $k_w$ estimated by $u_{10}$-based formulations tended to zero, with the exception of the formulations by McGillis et al. (2001) and Wanninkhof et al. (2009) as explained below. This is a bias from reality that has been thoroughly debated during the last decades. The COARE algorithm addressed it by adding a gustiness term to stabilize the $k_w$ in effective velocities under lighter winds (Grachev and Fairall, 1996; Fairall et al., 2003). With the same objective, Clayson et al. (1996) replaced the gustiness term by a capillary wave parameterization. Mackay and Yeun (1983), McGillis et al. (2001), Wanninkhof et al. (2009) and Johnson (2010) added a constant to the $k_w$ equation. In our case, due to the iWLP (equations 14 and 16) and the $k_w$ dependence on $u_*$, under the lowest winds but as long as there are waves, our comprehensive algorithms always provide effective velocities similar to the estimated by the authors mentioned above. Hence, our solution resembles the solution by Clayson et al. (1996). ~~These results highlight the potential of the SBL stability and the sea-surface agitation as additional $k_w$ mediators. It is curious that the wave variables were the responsible for the big differences between $k_w$ estimates (as shown in Fig. 3) although these were the variables to which the $k_w$ was least elastic (as shown in Fig. 4). It demonstrates that more important than model sensitivity (or elasticity) is how much the respective variables effectively change in the real world.i~~s yet interesting detail of how the WLLP and the iWLP diverged under smoother sea-surfaces (*not shown*), supporting the solution suggested in the COARE 3.0 (Fairall et al., 2003) for the iterative estimation of $u_*$ and $z_0$.

 Complementary to the analysis above, we also used timulations of the European costal oceans to compare between the ESM standard (the Wan92) and  our comprehensive alternatives. (We show the comparison with the iWLP ZRb0va). ~~chosen on the basis of two factors: it was both the most elastic formulation and the one providing the closest estimates to the Wan92 (recall Fig. 3). Since the Wan92 often represented the central tendency of the iWLP ZRb03, this choice provided the best probability that the differences between the $k_w$ estimates were due to the enhanced representation of the environmental processes involved and not to systematic biases associated to uncertainty in the parameter estimation. Bothunder two particularsituations (Fig. 5): (i)low winds and, and (ii) under high winds and rougher sea-surfaces~~(Fig. 5). The details of the simulations and the differences between $k_w$ estimates are presented hereafter.

Strong winds occurred along the European shores from the 24th to the 26th of May of 2014. Besides, the air was unusually cold for the season and colder than the sea-surface (Video 1). The rise of the warmer air, heated by the sea-surface, generated turbulent eddies that enhanced mixing within the SBL. These unstable conditions

were identified by $Ri_b<0$, L tending to $^-0$ and $\psi_m<0$ (Video 2). The mixing of the SBL enhanced $u_*$ and $k_w$ everywhere the wind blew lighter. This situation occurred more frequently and intensively nearby land masses and often associated to cooler continental breezes blowing off-shore. Its correct simulation required the estimation of the $Ri_b$, L and $\psi_m$ from the algorithms by Grachev and Fairall (1997) and Stull (1988) that account for humidity considering saturation at 0 m heights. The $Ri_b$ estimates neglecting humidity, following (Lee, (1997),  yielded biased estimates of the  SBL conditions(i.e, with $Ri_b>0$) as consequence of biased estimates of the virtual potential temperature. Stull (1988, page 9), Grachev and Fairall (1996) and Fairall et al. (2003) already highlighted the importance of accounting for humidity.

The sea-surface agitation was very heterogenic, particularly at the coastal ocean where it attained both the highest and the lowest estimated roughness lengths (the $z_0$ in Video 3). There, the steeper waves, consequence of shorter fetches, should extract more momentum from the atmosphere under similar $u_{10}$ conditions (Taylor and Yelland, 2001; Fairall et al, 2003). Thus, the rougher coastal ocean surfaces were expected to possess more turbulent layers through which gases were transferred at higher rates. The comprehensive formulations simulated this by increasing $u_*$ (and consequently $k_{wind}$) with $z_0$ under similar $u_z$ i.e, similar winds generate more drag when blowing over harsher sea-surfaces. Aside the rougher weather, whenever lighter wind blew over smoother sea-surfaces, the iWLP estimated much higher $z_0$ than the WLLP (video 4), demonstrating that the smooth flow was a fundamental driver for the $z_0$ under calmer weather. This increase in $z_0$ lead to significantly higher $u_*$, often 1.5 times higher and sometimes more, anticipating a significant impact on the $k_{wind}$ estimates.

In some rare situations the algorithms estimated unreasonably high $k_w$ despite the $z_0$ bounds imposed in the software. To avoid this bias, all $k_w$ estimates were imposed a 200 cm·h$^{-1}$ceiling.

The and the  formulations often diverged) their $k_w$ estimates, particularly in the coastal ocean, both on the Atlantic side and on the Mediterranean side, and mostly associated to storms (Video 5). Integrated over space and time, the Wan92 transferred 33061 km$^3$ of $CO_2$ across the $\approx$5,054,896 km$^2$ of ocean surface during the 66 h that the experiment lasted, corresponding to 90.8% of the 36392 km$^3$ of $CO_2$ transferred by the W05va formulation. However, as the bias occurred in both directions the absolute bias summed to 11880 km$^3$. These differences were higher at the coastal ocean (Fig. 6), a consequence of the factors that were not taken into consideration by the Wan92 (the ESM standard). Apart the $CO_2$, the W05va transferred 35479 Km$^3$ of $CH_4$ and of $N_2O$. ~~Its largest estimates of $k_w$ were associated to unreasonably high estimates of $z_u$ that biased the subsequent results. These biased estimates of $z_u$ could either be due to a poor calibration of the Taylor and Yelland (2001) model estimating $z_u$ from the wave field or due to biased wave field provided by the WW3-NEMO. To avoid this bias, $k_w$ was imposed a 70 cm·h$^{-1}$ceiling, corresponding to the maximum reported in the bulk literature associated to similar wind speeds. With this restriction, the difference in the $CO_2$ volume transferred by either formulation across the $\approx$5,054,896 km$^2$ of ocean surface during the 66 h was of 12997 km$^3$, corresponding to 33.7% of the 38551 km$^3$ of $CO_2$ total volume transferred using the ESM standard formulation (Fig. 6). These differences were higher at the coastal ocean, a consequence of the factors that were not taken into consideration by the ESM standard.The total~~

volumes of $CH_4$ and of $N_2O$ transferred were 41156 $Km^3$ and 41158 $Km^3$, respectively. Thatever the greenhouse gas, the differences were negligible between estimating $k_w$ using the single layer or the double layer scheme (Video 5).  Nevertheless, it is worth noting that it was again under the Mediterranean storms that the bigger differences were found.

**4    Discussion**

~~The accurate estimation of the balances of greenhouse gases and aerosols in the atmosphere and in the oceans, as well as their fluxes across the surfaces of the coastal oceans, is an important issue for biogeosciences and Earth-System modelling (ESM). Previous estimates of $CO_2$ uptake by the global oceans done by coarse resolution implementations diverged in about 70 % depending on the transfer velocity formulations being used (Takahashi et al., 2002), whereas the wide uncertainty in the ocean $N_2O$ source to the atmosphere mostly originated from the uncertainty in the air–water transfer velocities (Nevison et al., 1995). However, the knowledge on this subject is still limited, with plenty of room for improvement. As an example, the simpler formulations for the estimation of $k_w$ assume either a quadratic or cubic dependency from $u_{10}$ depending mostly on the sensing method, time scale and fetch at the particular location. Furthermore, the simulation of atmosphere-ocean interactions by regional and Earth-system models, by still using these simpler formulations, are decades behind the state-of-the-art. Our work proposes a framework to incorporate this state-of-the-art in an atmosphere-ocean coupler and demonstrates that this is fundamental for reliable simulations of coastal ocean systems.~~

Remarkably, oth solubility formulations generally estimated similar solubilities  despite their distinct chemistry backgrounds. Nevertheless, they did diverge in as much as 0.045 $mol \cdot mol^{-1}$ of $CO_2$, 0.0015 $mol \cdot mol^{-1}$ of $CH_4$ and 0.012 $mol \cdot mol^{-1}$ of $N_2O$ (i.e, mol of gas in the ocean surface per mol of gas in the atmosphere) in some of the most sensitive situations for Earth-System modelling and satellite data processing: (i) the cooler marine waters occur closer to the poles, where the solubility pump traps greenhouse gases and carries them to the deep ocean (Sarmiento and Gruber, 2013), and (ii) the warmer and the less saline waters occurring at the coastal ocean and seas, which have regularly been observed having greenhouse gases and DMS  dissolved in concentrations highly unbalanced with those of the atmosphere (Nevison et al., 2004; Borges et al, 2005; Barnes and Upstill-Goddard, 2011; Sarmiento and Gruber, 2013; Dutta et al., 2015; Gypens and Borges, 2015; Harley et al., 2015). Therefore, the biases in the estimated total amount of greenhouse gases in the first meter depth of the European coastal ocean during late May 2014 may be an indicator of higher global biases.

The accurate estimation of the transfer velocities of greenhouse gases and DMS across the ocean surface is a fundamental issue for biogeosciences and Earth-System modelling. Previous estimates of $CO_2$ uptake by the global oceans done by coarse resolution implementations diverged in about 70 % depending on the formulations being used (Takahashi et al., 2002), whereas the wide uncertainty in the ocean $N_2O$ source to the atmosphere mostly originated from the uncertainty in the air–water transfer velocities (Nevison et al., 1995). Despite its importance, the knowledge on this subject is still limited, with plenty of room for improvement. The simpler formulations assume either quadratic or cubic dependencies of $k_w$ from $u_{10}$ depending mostly on the sensing method, time scale and fetch (Wanninkhof, 1992; Nightingale et al., 2000; McGillis et al., 2001; Ho et al., 2006; Sweeney et al., 2007; Wanninkhof et al., 2009). Meanwhile, there were substantial developments on the effects of other factors as wave state, atmospheric stability,

currents, surfactants, rain and ice cover. Our framework integrates these factors and allows comparison among algorithms of different degrees of complexity. Furthermore, we programmed it to automatically select simpler algorithms when lacking variables indispensible for the application of the more comprehensive ones. Hence, this framework can also be used as basis for atmosphere-ocean couplers in regional and Earth-system models. Our comparisons demonstrated that the more comprehensive algorithms outrival the simpler ones by taking into consideration factors that are fundamental for  the transfer velocity of greenhouse gases and DMS across the coastal oceans' surface. The most determinant factors were the atmospheric stability of the SBL and the sea-surface roughnessSimilar conclusions were recently achieved by Jackson et al. (2012) and Shuiqing and Dongliang (2016).

The more comprehensive formulations still need improvement and validation. It is imperative to calibrate and validate the estimation of transfer velocity ($k_w$) from friction velocity ($u_*$) and wind-wave breaking ($k_{bubble}$), and the roughness length ($z_0$) from the wave field. All the available formulations for these specific purposes lack robust parameter estimations. Generally, there seems to be a great dependency of the available algorithms from the particular data sets that were used to calibrate them. Nevertheless, there is a general consensus that the $k_{bubble}$ term is fundamental under high wind speeds, with its estimate being central to current $k_w$ research. The latest developments have been on the dependency of $k_{bubble}$ from the interactions among the wind, the wave state, the bubble plume and the properties of the gas being transferred (Woolf et al. 2007; Callaghan et al., 2008, 2014; Goddijn-Murphy et al., 2011, Crosswell, 2015). The effect of sea-spray  recently became a prominent topic with the emergence of algorithms like the ones by Zhao et al. (2006) and Wu et al. (2015). and a dedicated section in the latest atmosphere-ocean interactions workshop in Brest. So far, these focused on the momentum transfer from wind to the ocean surface and the attenuation of the friction velocity. It should be interesting to understand how the intrusion of the sea-spray on the atmosphere affects the transfer velocity of gases, being anticipated a process symmetrical to that of the intrusion of bubbles on the ocean. The new algorithms for the effects of surfactants are particularly concerned with the variability of the coastal ocean (Pereira et al., 2016). These no longer associate the surfactants to the Schmidt number's exponent but rather to a coefficient setting a proportional decay of $k_w$. The effect of sea-ice must take into consideration its distortion of the ocean surface and its effect upon the SBL stability (Loose et al., 2014). Our coupling solution still needs to integrate the effects of the sea-surface cool-skin and warm-layer, surfactants, rain, sea-spray and sea-ice. From these, the cool-skin and warm-layer algorithms are the only with robust calibrations and validations, mostly done under the COARE (Fairall et al., 1996; Fairal et al., 2003; Zeng and Beljars, 2005; Brunke et al., 2008). The addition of complexity to any coupling solution must be carefully thought as these cannot become intricate to the point of computation becoming unbearable for ESM application. Algorithm making extensive use of for-loops are unviable (i) loops are computationally slow and (ii) can easily become conflictive with vectorization and its coordination with parallel processing. Hence our software needed a deep restructuration from its original version presented in Vieira et al. (2013). Once done, vectorization *per se* enabled improving calculations roughly 12× faster in a single core.

530

~~Besides finding the appropriate algorithms and parameter values to be used by the coupler, there is also the issue of accurately retrieving the variables mediating the gas transfer. The results showed that the $k_w$ was most elastic to the variables related with the SBL stability, namely the $u_{10}$, temperature, pressure and humidity. Although these are provided by the oceanic and atmospheric model components at courser vertical resolutions, they need to be transposed to finer vertical resolutions taking into consideration the processes occurring at the sea-surface. While the $u_{10}$ is given by the atmospheric model, the water temperature needs to account for the cool-skin and warm-layer effects and the heat and humidity at the SBL need to account for their vertical fluxes over the sea-surface.for these tasksE-C methods toeionthe fluxesacross the SBLerin its latterdid the COAREaddressedof gases and the importance of sea-surface roughnessJeffrey et al., 2010;Given~~Unfortunately, its complexity is deterrent of application to already computationally intensive geoscientific models. The transferred quantities of heat, moisture and momentum, their transfer coefficients, the dimensionless roughness lengths and the gustiness term are interdependent and must be solved iteratively. The COARE estimation of the $\psi_m$ alone is computationally heavier than the most comprehensive ensemble possible in the FuGas 2.2. The COARE was recently introduced as optional in the MOHID model and software. Our preliminary trials testing the heat transfer took $\approx 14\%$ longer to run each simulation.

**5    Code and Data Availability**

[revised manuscript text omitted]

OCMIP: Ocean Carbon-Cycle Model Intercomparison Project, available at: http://ocmip5.ipsl.
jussieu.fr/OCMIP, last updated: 2004 (last access: 27 August 2015), 2004.

Pereira, R., Scheneider-Zapp, K. and Upstill-Goddard, R.: Surfactant control of gas transfer velocity along an off-shore
coastal transect: results from a laboratory gas exchange tank. Biogeosciences Discuss., doi:10.5194/bg-2016-7,
2016.

Raymond, P. A. and Cole, J. J.: Gas exchange in rivers and estuaries: choosing a gas transfer velocity, Estuaries, 24,
312–317, 2001.

Rutgersson, A., Norman, M., Schneider, B., Petterson, H. and Sahlée, E.: The annual cycle of carbon dioxide and
parameters influencing the air–sea carbon exchange in the Baltic Proper. J. Mar. Sys., 74: 381-394, 2008.

Sander, R.: Compilation of Henry's law constants (version 4.0) for water as solvent. Atmos. Chem. Phys., 15, 4399-
4981, doi:10.5194/acp-15-4399-2015, 2015.

Sarmiento, J. L. and Gruber, N.:. *Ocean Biogeochemical Dynamics*. Princeton University Press, New Jersey, USA.
pp73-100, 2013.

Shuiqing, L. and Dongliang, Z.: Gas transfer velocity in the presence of wave breaking. Tellus B, 68, 27034, 2016.

Soloviev, A., Donelan, M., Graber, H., Haus, B. and Schlüssel, P.: An approach to estimation of near-surface turbulence
and CO2 transfer velocity from remote sensing data, J. Mar. Sys., 66, 182-194, 2007.

Stull, R. B.: An introduction to Boundary Layer Meteorology, Kluwer Academic Publishers, Dordrecht, pp151-195,
1988.

Sweeney, C., Gloor, E., Jacobson, A. R., Key, R. M., McKinley, G., Sarmiento, J. L. and Wanninkhof, R.: Constraining
global air-sea gas exchange for CO2 with recent bomb $^{14}$C measurements, Global Biogeochem. Cy., 21,
GB2015, doi:10.1029/2006GB002784, 2007.

Takahashi, T., Sutherland, S. C., Sweeney, C., Poisson, A., Metzl, N., Tilbrook, B., Bates, N.,Wanninkhof, R., Feelyf,
R. A., Sabine, C., Olafsson, J., and Nojirih, Y.: Global sea–air CO2 flux based on climatological surface ocean
$p$CO$_2$, and seasonal biological and temperature effects, Deep-Sea Res., 49, 1601–1622, 2002.

Taylor, P. K. and Yelland, M. J.: The dependence of sea surface roughness on the height and steepness of the waves, J.
Phys. Oceanogr., 31, 572–590. 2001.

Vieira, V. M. N. C. S., Martins, F., Silva, J. and Santos, R.: Numerical tools to estimate the flux of a gas across the air–
water interface and assess the heterogeneity of its forcing functions. Ocean Sci., 9, 355-375, 2013.

Wang, B., Liao, Q., Fillingham, J. and Bootsma, H. A.: On the coefficients of small eddy and surface divergence
models for the air-water gas transfer velocity, J. Geophys. Res. Oceans, 120(3), DOI: 10.1002/2014JC010253,
2015.

Wanninkhof, R.:Relationship between wind speed and gas exchange over the ocean, J. Geophys. Res., 97, 7373–7382,
1992.

Wanninkhof, R., Asher, W. E., Ho, D. T., Sweeney, C. S., and McGillis, W. R.: Advances in quantifying air-sea gas
exchange and environmental forcing, Ann. Rev. Mar. Sci., 1, 213–244,
doi:10.1146/annurev.marine.010908.163742, 2009.

Webb, E. K., Pearman, G. I. and Leuning, R.: Correction of flux measurements for density effects due to heat and water vapour transfer., Quart. J.R. Meteorol. Soc., 106: 85-100, 1980.

Weiss, R. F.: Carbon dioxide in water and seawater: the solubility of a non-ideal gas., Mar. Chem., 2, 203-215, 1974.

Weiss, R. F. and Price B. A.: Nitrous oxide solubility in water and seawater., Mar. Chem., 8, 347-359, 1980.

Woolf, D. K.: Parameterization of gas transfer velocities and sea state-dependent wave breaking,Tellus B, 57, 87 – 94, 2005.

Woolf, D. K., Leifer, I. Nightingale, P.D., Andreae, M.O.: Modelling of bubble-mediated gas transfer: Fundamental principles and a laboratory test., J. Mar. Sys., doi: 10.1016/j.jmarsys.2006.02.011, 2007.

Wu, L., Rutgersson, A. and Sahlée, E.: The impact of waves and sea-spray on modelling storm track and development., Tellus A, 67, 27967, 2015.

Zeng, X. and Beljaars, A.: A prognostic scheme of sea surface skin temperature for modelling and data assimilation. Geophys. Res. Lett., 32, L14605, 2005.

Zhang, W., Perrie, W. and Vagle, S.: Impacts of winter storms on air-sea gas exchange. Geophys. Res. Lett., 33, L14803, 2006.

Zhao, D., Toba, Y., Suzuki, Y., and Komori, S.: Effect of wind waves on air-sea gas exchange: proposal of an overall $CO_2$ transfer velocity formula as a function of breaking-wave parameter, Tellus B, 55, 478–487, 2003.

Zhao, D., Toba, Y., Sugioka, K. and Komori, S.: New sea spray generation function for spume droplets. J. Geophys. Res., 111(C2), doi: 10.1029/2005JC002960, 2006.

8    **Figures**

[Figure]

**Figure 1:** Comparing solubility formulations

[Figure]

725

**Figure 2: Bias in the gas mass balance for the European coastal ocean**

[Figure]

730

**Figure 3**: **Comparing transfer velocity algorithms using**

735

[Figure]

740

**Figure 4**: **Elasticities of the transfer velocity to the environmental variables**.

[Figure]

[Figure]

[Figure]

[Figure]

**Figure 5: Applying the modelled data about the European coastal ocean for a direct comparison between the $k_w$ estimates**.

[Figure]

**Figure 6: Comparing transfer velocity algorithms using modelled data**

**Formatada:** Tipo de letra: (predefinido) Times New Roman

**9      Figure legends**

**Figure 1: Comparing solubility formulations**: match-mismatch between formulations estimating solubilities of the greenhouse gases $CO_2$, $CH_4$ and $N_2O$ according to water temperature ($T_w$), salinity (S) and location. Colorscale is ($k_H$) Henry's constant estimated from (Joh10) Johnson, 2010, or (Sar13) Sarmiento and Gruber, 2013. Colorscale: quotient between $k_H$ estimated from Johnson, 2010 ( $k_{H'Joh10'}$) and $k_H$ estimated from Sarmiento and Gruber, 2013 ($k_{H'Sar13'}$) i.e, $k_{H'Joh10'}/k_{H'Sar13'}$.

**Figure 2: Bias in the gas mass balance for the European coastal ocean**: comparing algorithm by Johnson (2010) to compilation by Sarmiento and Gruber (2013). Colorscale: $\Delta$ton $\cdot m^{-1} \cdot 121$ $km^{-2}$ i.e, bias in the gas mass estimated by each algorithm ($\Delta$ton) for the first meter depth ($m^{-1}$) in 11 km wide cells (121 $km^{-2}$).

**Figure 3: Comparing transfer velocity algorithms using the data observed at the Baltic**. (a) sea-surface roughness given by significant wave height ($H_s$) and peak wave period ($L_p$). (b) tThe $k_w$ estimated by renowned $u_{10}$-based formulations (lines)and by some of the most comprehensive alternatives provided in the FuGas 2.1, usingcompared to the $k_w$ estimated datfroma the Eddy-Covariance measurements (circles) observed at the Baltic. (c) comparing the $k_w$ estimated by $u_{10}$-based formulations (lines) and by comprehensive alternatives (circles). Simple formulations by 'Wan92' - Wannninkhof (1992),: 'WMG99' – Wanninkhof and McGillis (1999), 'We09' – Wanninkhof et al. (2009),: 'Sea07' – Sweeney et al. (2007), 'Nea00' – Nigthingale et al. (2000),: 'McG01' – McGillis et al. (2001), 'Ho+06' – upper boundary in Ho et al. (2006) 'Ho-06' – lower boundary in Ho et al. (2006) . Comprehensive formulations were: assembled using the 'iWLP' – iteratively estimated wind log-linear profile and included the 'Jea87' – Jähne et al

(1987).; 'Zhg06' - Zhang et al. (2006); 'ZRb03' - Zhao et al (2003). and; 'W05av' – Woolf (2005) with the kinematic viscosity of air.;

assembled using the 'WLLP' - wind log-linear profile or the 'iWLP' - iteratively estimated wind log-linear profile.

**Figure 4: Elasticities of the transfer velocity to the environmental variablesits forcing functions.** Elasticities $(\partial k_w/k_w)/(\partial x/x)$ estimated using the data observed at the Baltic. The $k_w$ was estimated by the iterative wind log-linear profile (iWLP) with the Zhao et al (2003) $k_{bubble}$ term (ZRb03) for the 60 observations in the Baltic. The box-and-wiskers represent the quartiles.

**Figure 5: Applying the modelled data about the European coastal ocean for a direct comparison between the $k_w$ estimates** by the ESM standard –Wan92 - and our best performinga comprehensive formulation - W05va - including the $k_{bubble}$ term by Woolf (2005), the $k_{wind}$ term by Jahne et al. (1987), the $z_0$ term from the COARE 3.0 and the iterative wind log-linear profile. The $z_0$ is given in m and the $u_*$ in $m \cdot s^{-1}$.

**Figure 6: Comparing transfer velocity algorithms using modelled data**: ($k_w$ $CO_2$ Wan92) transfer velocity of $CO_2$ estimated from the formulation by Wanninkhof (1992) and averaged over the 66 h; by the 'iWLP' - iterative Wind Log-Linear Profile and the 'ZRb03' - Zhao et al. (2003) formulation, ($\Delta k_w$ $CO_2$) difference between the iWLP with ZRb03 and the Wan92' - Wanninkhof (1992) formulation, (NRMSE) Normalize Root Mean Square Error between estimating the transfer velocity using the formulation by Wanninkhof (1992) or the formulation by Woolf (2005) conjugated with the iterative wind log-linear profile. iWLP with ZRb03 and the Wan92, , ($\Delta k_w$ $CH_4$) difference between the single and double layer schemes using the iWLP with ZRb03. Colour scale: volume (or $\Delta$ volume) transferred in units of $Km^3/66h$, except for NRMSE.

---

## Author Comment (AC4) · 9 Jan 2017

**The FuGas 2.2 framework for atmosphere-ocean coupling in geoscientific models: comparing and improving algorithms for the estimates of the solubilities and fluxes of greenhouse gases and DMS**

[revised manuscript text omitted]

$$k_H = k_{H,0} \cdot 10^{K_s S} \tag{7}$$

$$\theta = 7.33532 \cdot 10^{-4} + 3.39615 * 10^{-5} \cdot \log(k_{H\#})$$
$$-2.40888 \cdot 10^{-6} \cdot \log(k_{H\#})^2 \tag{8}$$
$$+1.57114 \cdot 10^{-7} \cdot \log(k_{H\#})^3$$

130

The mismatches between both algorithms lead to differences in the estimates of greenhouse gases dissolved in the first meter below the ocean surface, which were calculated from $\Delta \text{ton} \cdot \text{m}^{-1} \cdot 121 \text{ km}^{-2} = 11^2 \cdot \Delta s \cdot p_{gas} \cdot P \cdot 101325 \cdot M_a/(10^9 \cdot R \cdot T)$. The $\Delta s$ was the difference in the solubilities estimated by both algorithms and converted to the $C_w/C_a$ form. Hence,

135 $\Delta s = 1/k_{H'Sar13'} - 1/k_{H'Joh10'}$, and because $C_a$ was equal among them, the $\Delta s = (C_{w'Joh10'} - C_{w'Sar13'})/C_a$. This difference of mol·m$^{-3}$ of gas in the water per mol·m$^{-3}$ of gas in the air at each cell was averaged over the 66 h time interval. In order to convert from mols to grams in the water we multiplied by the molar mass of the specific gas ($M_g$), which is 44.01 for $CO_2$, 16.043 for $CH_4$ and 44.013 for $N_2O$. Then, we divided by $10^6$ to convert from grams to tons. We still needed to estimate $C_a$ from the atmospheric pressure (P) and the partial pressure ($p_{gas}$) of $CO_2$, $CH_4$ or $N_2O$, 390 ppm, 1.75 ppm

140 and 0.325 ppm respectively (EPA, 2015), assuming that they were approximately uniform all over the atmospheric surface boundary layer (SBL). Using the ideal gas law, we divided by R and T (in Kelvin), multiplied by 101325 to convert Pascal to atm and divided by $10^6$ to re-scale from ppm to unity. Finally, we multiplied by (11 km)$^2$ to have the total difference in mass dissolved in the first meter below the surface of 11 km wide cells.

**2.2    Transfer velocity**

145    The available algorithms consider that the rate at which gases cross the sea-surface is basically set by the turbulence upon it. E.g. wind drag, wave breaking, currents and rain promote turbulence. The water viscosity, set by temperature and salinity and enhanced by the presence of surfactants, antagonizes turbulence. Figure 1 in the work by Wanninkhof et al. (2009) clarifies how some of these processes interact.  With all these forcings, it becomes difficult to develop an algorithm that estimates the transfer velocity accurately. The literature has many of them, either fitted to specific surface

150    conditions or rougher generalizations, focusing on different factors and relying in different theoretical backgrounds. The simpler ones rely on the wind velocity 10m above the sea-surface ($u_{10}$). Among then, the formulation by Wanninkhof (1992) (henceforth also mentioned as 'Wan92') became the standard used in ESM and satellite data processing (equation 9a,b).  It further considers the Schmidt number of the water ($Sc_w$) related to viscosity and with its exponent reflecting the surface layer's rate of turbulent renewal. Under low winds, the transfer velocity of $CO_2$ is chemically

155    enhanced due to reaction with water ($\alpha_{Ch}$) and scales with temperature.

$$k_w = (\alpha_{Ch} + 0.31 \cdot u_{10}^2) \left(\frac{Sc_w}{660}\right)^{-0.5} \tag{9a}$$

$$\alpha_{Ch} = 2.5 \cdot (0.5246 + 0.0162 T_w + 0.000499 T_w^2) \tag{9b}$$

160    Other simple empirical formulations based only on $u_{10}$ (Carini et al., 1996; Raymond and Cole, 2001), or also accounting for current drag with the bottom (Borges et al., 2004), used data collected in estuaries under low wind conditions. However, modelling the coastal ocean at finer resolutions requires an enhanced representation of the multitude of processes involved. Hence, we updated the framework by Vieira et al. (2013), with the $k_w$ being decomposed into its shear produced turbulence ($k_{wind}$) and bubbles from whitecapping ($k_{bubble}$) forcings (Asher and

165    Farley, 1995; Borges et al, 2004; Woolf, 2005; Zhang et al., 2006). $Sc_w$ was determined from temperature and salinity following Johnson (2010):

$$k_w = (\alpha_{Ch} + k_{bubble} + k_{wind} + k_{current}) \cdot (600/Sc_w)^{0.5} \tag{10}$$

170    The formulation by Zhao et al. (2003), merged $k_{wind}$ into $k_{bubble}$ (Eq. 11a) using the wave breaking parameter ($R_B$ given by Eq. 11b). The $u_*$ is the friction velocity i.e, the velocity of wind dragging on the sea-surface, and $f_p$ is the peak angular frequency of the wind-waves. The kinematic viscosity of air ($\nu_a$) was estimated from Johnson (2010). This solution used the wave field as a proxy for whitecapping that increased transfer velocity with wind-wave age. However, it simultaneously used the wave field as a proxy for the sea-surface roughness that increased transfer velocity from

175    wind-drag over steeper younger waves (through the WLLP estimation of $u_*$ explained in a section below).

$$k_{bubble} = 0.1315 \cdot R_B^{0.6322} \tag{11a}$$

$$R_B = \frac{u_*^2}{2\pi f_p \nu_a} \tag{11b}$$

180    A more comprehensive solution split the two drives of transfer velocity (Woolf, 2005; Zhang et al., 2006): $k_{wind}$ for the transfer mediated by the turbulence generated by wind drag (Eq. 12, taken from Jähne et al., 1987) and $k_{bubble}$ for the

transfer mediated by the bubbles generated by breaking waves (Eq. 13). B is Bunsen's solubility coefficient estimated for the local sea-surface conditions. $W=3.88\times10^{-7}R_B^{1.09}$ is the whitecap cover requiring the $R_B$ estimated from Eq. (11b), V=4900, e=14 and n=1.2.

185

$$k_{\text{wind}} = 1.57 \cdot 10^{-4} \cdot u_* \tag{12}$$

$$k_{\text{bubble}} = \frac{WV}{B}\left[1 + \left(e \cdot B \cdot Sc_{w^{-1/2}}\right)^{-1/n}\right]^{-n} \tag{13}$$

These formulations required friction velocity ($u_*$), which was estimated from the Wind Log-Linear Profile (WLLP: Eq.

190 14) accounting for wind speed at height z ($u_z$), atmospheric stability of the surface boundary layer (through $\psi_m$) and sea-surface roughness (through the roughness length $z_0$). The $\kappa$ is von Kármàn's constant. Historically, the WLLP originated from the Monin-Obukhov Similarity Theory (Monin and Obukhov, 1954; Stull, 1988).

$$u_* = \frac{u_z \cdot \kappa}{\ln(z) - \ln(z_0) + \psi_m(z, z_0, L)} \tag{14}$$

195

Roughness length ($z_0$) is the theoretical minimal height (most often sub-millimetrical) at which wind speed averages zero. It is dependent on surface roughness and often used as its index. It is more difficult to determine over water than over land as there is a strong bidirectional interaction between wind and sea-surface roughness. Taylor and Yelland (2001) proposed a dimensionless $z_0$ dependency from the wave field, increasing with the wave slope (Eq. 15). Here, $H_s$

200 is the significant wave height and $L_p$ is the peak wave period. Due to the bidirectional nature of the $z_0$ and $u_*$ relation, we also tested an iterative solution (iWLP) where Eq.15 was used as a first guess for the $z_0$ and Eq.14 for its subsequent $u_*$. A second iteration re-estimated $z_0$ from the COARE 3.0 (Fairall et al.; 2003) adaptation of the Taylor and Yelland (2001) formulation, which added a term for smooth flow (Eq. 16), and $u_*$ again from Eq.14. Applying four iterations were enough for an excellent convergence of the full data array. Irrespective of the WLLP or iWLP algorithm, the

205 coefficients proposed by Taylor and Yelland (2001) applied to our data sometimes yielded incredibly high and unreal $z_0$ leading to absurdly high $u_*$ and $k_w$. To prevent this bias we imposed a maximum roughness length $z_{0,max}=0.01$ m.

$$\frac{z_0}{H_s} = 1200 \cdot \left(\frac{H_s}{L_p}\right)^{4.5} \tag{15}$$

$$z_0 = 1200 \cdot H_s \left(\frac{H_s}{L_p}\right)^{4.5} + \frac{0.11\nu_a}{u_*} \tag{16}$$

210

Atmospheric stability characterized the tendency of the surface boundary layer (SBL) to be well mixed (unstable SBL with $\psi_m<0$) or stratified (stable SBL with $\psi_m>0$). The $\psi_m$ was inferred from the 'bulk Richardson number' ($Ri_b$: Eq. 17), weighting the air vertical heat gradient and kinetic energy, and requiring the air virtual potential temperature ($T_v$) estimated from air temperature (T in ºC), air pressure (P in atm), humidity (dimensionless) and the gravitational

215 acceleration constant (g). Grachev and Fairall (1997) estimated $T_v=T_p(1+0.61q)$, where $T_p$ is the air potential temperature and q the observed specific humidity . Stull (1988) estimated $T_v=T_p(1+0.61\cdot r_{sat}\cdot h_r+r_L)$, where $r_{sat}$ is the water vapour mixing ratio at saturation, $h_r$ is the observed relative humidity and $r_L$ the observed liquid water mixing ratio. $T_p$ (in ºC) was estimated from $T_p = T_k(1000/(1013.25P))^{0.284}$, where $T_k$ is temperature (Kelvin). The $r_{sat} =$

$0.622e_{sat}/(101.32501P-e_{sat})$ and the $\ln(e_{sat}) = \ln0.61078 +17.2694T/(T_k-35.86)$. Alternatively, Lee (1997) estimated the $Ri_b$ directly from the air potential temperature neglecting humidity. The wind velocity ($u_z$), temperature ($T_z$), pressure ($P_z$) and humidity ($q_z$) z meters above sea-surface were given by the WRF second level. The wind velocity at $z_0$ ($u_0$) was set to the theoretical $u_0$=0. Temperature at the height of 0 m ($T_0$) was given by the SST (Grachev and Fairall, 1997; Fairal et al., 2003; Brunke et al., 2008) without rectification for cool-skin and warm-layer effects due to the lack of some required variables. Yet, these effects tend to compensate each other (Brunke et al., 2008; Fairall et al., 1996; Zeng and Beljars, 2005). Air pressure at 0 m ($P_0$) was given by the WRF at the lower first level (at roughly 0 m). Humidity at 0 m ($q_0$) was set to the saturation level at $P_0$ and $T_0$ using $q=r_{sat}/(1+r_{sat})$ (Grachev and Fairall, 1997). The $Ri_b$ was used to estimate the length L from the Monin-Obukhov's similarity theory, a discontinuous exponential function tending to $\pm\infty$ when $Ri_b$ tends to $\pm0$ and tending to $\pm0$ when $Ri_b$ tends to $\pm\infty$. $Ri_b$ and L were used to estimate $\psi_m$ following Stull (1988) or Lee (1997) algorithms.

$$Ri_b = \frac{g\Delta T_v \Delta z_i}{T_v \cdot u_z^2} \tag{17}$$

The transfer velocity of mildly soluble gases, besides taking into consideration the molecular crossing of the water-side surface layer, should also take into consideration the molecular crossing of the air-side surface layer (Liss and Slater, 1974, Wanninkhof et al., 2009; Johnson, 2010). We compared between the use of the traditional single layer model and the double layer "thin film" model (Liss and Slater, 1974), the later estimating the air-side transfer velocity ($k_a$) from the COARE formulation as in Eq. 18 (Jeffrey et al., 2010). CD is the drag coefficient and $Sc_a$ the Schmidt number of air, which were determined for a given temperature and salinity following Johnson (2010).

[revised manuscript text omitted]

(i)   invoking the parallel processing toolbox was time consuming,

(ii)  replicating over 1Gb ram was time consuming,

(iii) when running the calculations with other programs on the background the 4Gb ram memory was soon exhausted. Thereafter, the use of virtual memory in the hard-disk stalled enormously the calculations. To prevent it the following actions were implemented,

(iv)  Programs like the antivirus, backup tools, Office, Skype, Dropbox, etc were shut down in the Task Manager,

(v)   the spmd were split into several sequential code blocks and in-between the variables no longer necessary were deleted. This spmd fragmentation was time consuming.

In conclusion, there is no perfect solution for parallel computing, and although spmd is the best strategy available for this task, its application needs to be carefully programmed according to the data and hardware characteristics. Optimizing the data management was one fundamental improvement between the 2.1 and the 2.2 versions of the FuGas. The simulations of the European coastal ocean no longer exhausted the ram memory and did not even use more than 70% of it (using above 90% becomes critical). Consequently, the computation time improved from ≈12 min to ≈4 min.

**3    Results**

The solubility formulations were compared simulating the range of environmental conditions commonly found in nature: $T_w$ ranged from 4ºC to 30ºC at 1ºC intervals while S ranged from 0 ppt to 36 ppt at 1 ppt intervals. The ratio between the solubilities estimated by each formulation (i.e, $k_{H,Joh10}/k_{H,Sar13}$) showed better how much these could diverge (Fig. 1). Afterwards, both formulations were applied to the conditions observed at the European coastal ocean during the experiment. The estimated solubilities were compared from their ratio averaged over the 66 h time interval using the geometric mean (Fig. 1). From the 24[th] to the 26[th] May the water temperature at the ocean surface changed significantly and there were large fresh water inputs from the Black Sea and the Baltic Sea (Video 1). The widest divergences were up to 4.5% in the $CO_2$ solubility estimates associated to cooler waters, 5.8% in the $CH_4$ solubility estimates associated to both temperature extremes, and 2.1% in the $N_2O$ solubility estimates associated to cooler and less saline waters (Fig. 1).These mismatches lead to large differences in the estimates of greenhouse gases dissolved in the first meter below the ocean surface (Fig. 2). Integrated over space, these differences summed to $1.17\times10^5$ ton of $CO_2$, 7374.5 ton of $CH_4$ and 25.1 ton of $N_2O$.

During the Baltic Sea sampling at the Östergarnsholm site the observed Δppm varied within 120 and 270, well below the limit for a 25% error in the flux estimates as reported by Blomquist et al. (2014) for our IRGA model, the LI-COR LI-7500. Even so, the $k_w$ estimated from the Eddy-Covariance (E-C) measurements were close matches to the $k_w$ estimated by both generalistic and comprehensive algorithms (Fig.3). The mismatches found in previous versions of this report were consequence of humidity mistakenly input in incorrect units into the E-C matlab script. The mismatches currently seen at lower wind speeds may be due to failure of the E-C measurements under atmospherically stable conditions, a problem well known to affect E-C methods. During the experiment the SBL was generally stable ($0<Ri_b<0.09$ with a couple of exceptions above 0.09) while the sea-surface was little to moderately rough ($z_0<0.49$ mm). These conditions were used as reference to estimate the elasticity of $k_w$ to the environmental variables (Fig. 4). The variables related with the SBL stability, namely the $u_{10}$, temperature, pressure and humidity, were able to induce larger changes in $k_w$. However, the SBL stability changed little during this experiment whereas the sea-state change considerably, with a calmer period during which the sea-surface was smoother and a harsher period during which the sea-surface was rougher (Fig.3). Hence, during this experiment the sea-state had a greater impact on the $k_w$ than the atmospheric stability. The $k_w$ dependency on the sea-state is well-known as it is thought that $k_w$ scales with the turbulent kinetic energy dissipation at the sea-surface ($\varepsilon$) and that this is better reflected by the sea-state (Soloviev et al., 2007, Wang et al., 2015). Accordingly, the COARE 3.0 included the wave state in the estimation of the roughness parameters essential for the transfer of mass, heat and momentum (Fairal et al. 2003) while Frew et al. (2004) observed a remarkable correlation between the $k_w$ and small scale waves. Our comprehensive algorithms adjusted to the sea-state splitting the $k_w$ estimates into two distinct groups relative to each period. The $k_w$ estimated for the rougher sea-states scattered along a steeper line placed above the $k_w$ estimated for the smoother sea-states. The $k_w$ estimated from the E-C measurements tended to follow this same pattern (Fig.3), and episodic departure from it may be inherent to E-C natural variability. The generalistic $u_{10}$-based formulations were unable to perform this adjustment to the local wave state. Their small $k_w$ fluctuations were a sole consequence of changes in water viscosity (as estimated by the $Sc_w$) driven by changes in water temperature. Furthermore, the Wan92 formulation and our comprehensive formulations adjusted remarkably well under rougher seas, whereas the Cea96 formulation calibrated with data from the Parker river estuary and our comprehensive formulations adjusted remarkably well under lighter winds and/or smoother seas (Fig.3). These fits

335 clearly show an ability of our comprehensive algorithms to adjust to the local conditions that cannot be met by the generalistic $u_{10}$-based formulations. This is not a minor detail: at $u_{10} \approx 8$ m·s$^{-1}$ the $k_w$ estimated from wind dragging over rougher or over smoother sea-surfaces differed $\approx 31\%$ while at $u_{10} < 4$ m·s$^{-1}$ the $u_{10}$-based formulations estimated less than 50% of the $k_w$ estimated by our comprehensive algorithms. Furthermore, under the lowest winds the $k_w$ estimated by $u_{10}$-based formulations tended to zero, with the exception of the formulations by McGillis et al. (2001) and Wanninkhof et

340 al. (2009) as explained below. This is a bias from reality that has been thoroughly debated during the last decades. The COARE algorithm addressed it by adding a gustiness term to stabilize the $k_w$ in effective velocities under lighter winds (Grachev and Fairall, 1996; Fairall et al., 2003). With the same objective, Clayson et al. (1996) replaced the gustiness term by a capillary wave parameterization. Mackay and Yeun (1983), McGillis et al. (2001), Wanninkhof et al. (2009) and Johnson (2010) added a constant to the $k_w$ equation. In our case, due to the iWLP (equations 14 and 16) and the $k_w$

345 dependence on $u_*$, under the lowest winds but as long as there are waves, our comprehensive algorithms always provide effective velocities similar to the estimated by the authors mentioned above. Hence, our solution resembles the solution by Clayson et al. (1996). There was yet the interesting detail of how the WLLP and the iWLP diverged under smoother sea-surfaces (*not shown*), supporting the solution suggested in the COARE 3.0 (Fairall et al., 2003) for the iterative estimation of $u_*$ and $z_0$.

350 We performed simulations of the European costal oceans to compare between the ESM standard (the Wan92) and our comprehensive alternatives. We show the comparison with the iWLP-W05va. Their $k_w$ estimates diverged mostly under unstable SBL, very rough or very smooth sea-surfaces, or higher friction velocities (Fig. 5). The details of the simulations and the differences between $k_w$ estimates are presented hereafter.

Strong winds occurred along the European shores from the 24$^{th}$ to the 26$^{th}$ of May of 2014. Besides, the air was
355 unusually cold for the season and colder than the sea-surface (Video 1). The rise of the warmer air, heated by the sea-surface, generated turbulent eddies that enhanced mixing within the SBL. These unstable conditions were identified by $Ri_b < 0$, L tending to $^-0$ and $\psi_m < 0$ (Video 2). The mixing of the SBL enhanced $u_*$ and $k_w$ everywhere the wind blew lighter. This situation occurred more frequently and intensively nearby land masses and often associated to cooler continental breezes blowing off-shore. Its correct simulation required the estimation of the $Ri_b$, L and $\psi_m$ from the
360 algorithms by Grachev and Fairall (1997) and Stull (1988) that account for humidity considering saturation at 0 m heights. The $Ri_b$ estimates neglecting humidity, following Lee (1997), yielded biased estimates of the SBL conditions as consequence of biased estimates of the virtual potential temperature. Stull (1988, page 9), Grachev and Fairall (1996) and Fairall et al. (2003) already highlighted the importance of accounting for humidity.

The sea-surface agitation was very heterogenic, particularly at the coastal ocean where it attained both the highest
365 and the lowest estimated roughness lengths (the $z_0$ in Video 3). There, the steeper waves, consequence of shorter fetches, should extract more momentum from the atmosphere under similar $u_{10}$ conditions (Taylor and Yelland, 2001; Fairall et al, 2003). Thus, the rougher coastal ocean surfaces were expected to possess more turbulent layers through which gases were transferred at higher rates. The comprehensive formulations simulated this by increasing $u_*$ (and consequently $k_{wind}$) with $z_0$ under similar $u_z$ i.e, similar winds generate more drag when blowing over harsher sea-
370 surfaces. Aside the rougher weather, whenever lighter wind blew over smoother sea-surfaces, the iWLP estimated much higher $z_0$ than the WLLP (video 4), demonstrating that the smooth flow was a fundamental driver for the $z_0$ under calmer weather. This increase in $z_0$ lead to significantly higher $u_*$, often 1.5 times higher and sometimes more, anticipating a significant impact on the $k_{wind}$ estimates.

In some rare situations the algorithms estimated unreasonably high $k_w$ despite the $z_0$ bounds imposed in the

375    software. To avoid this bias, all $k_w$ estimates were imposed a 200 cm·h$^{-1}$ceiling. The W05va and the Wan92

formulations often diverged their $k_w$ estimates, particularly in the coastal ocean, both on the Atlantic side and on the

Mediterranean side, and mostly associated to storms (Video 5). Integrated over space and time, the Wan92 transferred

33061 km$^3$ of $CO_2$ across the ≈5,054,896 km$^2$ of ocean surface during the 66 h that the experiment lasted, corresponding

to 90.8% of the 36392 km$^3$ of $CO_2$ transferred by the W05va formulation. However, as the bias occurred in both

380    directions the absolute bias summed to 11880 km$^3$. These differences were higher at the coastal ocean (Fig. 6), a

consequence of the factors that were not taken into consideration by the Wan92 (the ESM standard). Apart the $CO_2$, the

W05va transferred 35479 Km$^3$ of $CH_4$ and of $N_2O$. This formulation was also used to compare between the $k_{wind}$ and

$k_{bubble}$ components of $k_w$. The results showed that the $k_{bubble}$ term was always lower than the $k_{wind}$ term and only close to

it in two situations: (i) often in the fetch-unlimited Atlantic, and (ii) in a few storms inside the Atlantic where, given

385    their high winds, fetch was not a limitation. Whatever the greenhouse gas, the differences were negligible between

estimating $k_w$ using the single layer or the double layer schemes (Video 5). Nevertheless, it is worth noting that it was

again under the Mediterranean storms that the bigger differences were found.

**4       Discussion**

390    Both solubility formulations generally estimated similar solubilities despite their distinct chemistry backgrounds.

Nevertheless, they did diverge in as much as 0.045 mol·mol$^{-1}$ of $CO_2$, 0.0015 mol·mol$^{-1}$ of $CH_4$ and 0.012 mol·mol$^{-1}$ of

$N_2O$ (i.e, mol of gas in the ocean surface per mol of gas in the atmosphere) in some of the most sensitive situations for

Earth-System modelling and satellite data processing: (i) the cooler marine waters occur closer to the poles, where the

solubility pump traps greenhouse gases and carries them to the deep ocean (Sarmiento and Gruber, 2013), and (ii) the

395    warmer and the less saline waters occurring at the coastal ocean and seas, which have regularly been observed having

greenhouse gases and DMS dissolved in concentrations highly unbalanced with those of the atmosphere (Nevison et al.,

2004; Borges et al, 2005; Barnes and Upstill-Goddard, 2011; Sarmiento and Gruber, 2013; Dutta et al., 2015; Gypens

and Borges, 2015; Harley et al., 2015). Therefore, the biases in the estimated total amount of greenhouse gases in the

first meter depth of the European coastal ocean during late May 2014 may be an indicator of higher global biases.

400    The accurate estimation of the transfer velocities of greenhouse gases and DMS across the ocean surface is a

fundamental issue for biogeosciences and Earth-System modelling. Previous estimates of $CO_2$ uptake by the global

oceans done by coarse resolution implementations diverged in about 70 % depending on the formulations being used

(Takahashi et al., 2002), whereas the wide uncertainty in the ocean $N_2O$ source to the atmosphere mostly originated

from the uncertainty in the air–water transfer velocities (Nevison et al., 1995). Despite its importance, the knowledge on

405    this subject is still limited, with plenty of room for improvement. The simpler formulations assume either quadratic or

cubic dependencies of $k_w$ from $u_{10}$ depending mostly on the sensing method, time scale and fetch (Wanninkhof, 1992;

Nightingale et al., 2000; McGillis et al., 2001; Ho et al., 2006; Sweeney et al., 2007; Wanninkhof et al., 2009).

Meanwhile, there were substantial developments on the effects of other factors as wave state, atmospheric stability,

currents, surfactants, rain and ice cover. Our framework integrates these factors and allows comparison among

410    algorithms of different degrees of complexity. Furthermore, we programmed it to automatically select simpler

algorithms when lacking variables indispensable for the application of the more comprehensive ones. Hence, this

framework can also be used as basis for atmosphere-ocean couplers in regional and Earth-system models. Our

comparisons demonstrated that the more comprehensive algorithms outrival the simpler ones by taking into consideration factors that are fundamental for the transfer velocity of greenhouse gases and DMS across the coastal oceans' surface. The most determinant factors were the atmospheric stability of the SBL and the sea-surface roughness. Similar conclusions were recently achieved by Jackson et al. (2012) and Shuiqing and Dongliang (2016).

The more comprehensive formulations still need improvement and validation. It is imperative to calibrate and validate the estimation of transfer velocity ($k_w$) from friction velocity ($u_*$) and wind-wave breaking ($k_{bubble}$), and the roughness length ($z_0$) from the wave field. All the available formulations for these specific purposes lack robust parameter estimations. Generally, there seems to be a great dependency of the available algorithms from the particular data sets that were used to calibrate them. Nevertheless, there is a general consensus that the $k_{bubble}$ term is fundamental under high wind speeds, with its estimate being central to current $k_w$ research. The latest developments have been on the dependency of $k_{bubble}$ from the interactions among the wind, the wave state, the bubble plume and the properties of the gas being transferred (Woolf et al. 2007; Callaghan et al., 2008, 2014; Goddijn-Murphy et al., 2011, Crosswell, 2015). The effect of sea-spray recently became a prominent topic with the emergence of algorithms like the ones by Zhao et al. (2006) and Wu et al. (2015), and a dedicated section in the latest atmosphere-ocean interactions workshop in Brest. So far, these focused on the momentum transfer from wind to the ocean surface and the attenuation of the friction velocity. It should be interesting to understand how the intrusion of the sea-spray on the atmosphere affects the transfer velocity of gases, being anticipated a process symmetrical to that of the intrusion of bubbles on the ocean. The new algorithms for the effects of surfactants are particularly concerned with the variability of the coastal ocean (Pereira et al., 2016). These no longer associate the surfactants to the Schmidt number's exponent but rather to a coefficient setting a proportional decay of $k_w$. The effect of sea-ice must take into consideration its distortion of the ocean surface and its effect upon the SBL stability (Loose et al., 2014). Our coupling solution still needs to integrate the effects of the sea-surface cool-skin and warm-layer, surfactants, rain, sea-spray and sea-ice. From these, the cool-skin and warm-layer algorithms are the only with robust calibrations and validations, mostly done under the COARE (Fairall et al., 1996; Fairal et al., 2003; Zeng and Beljars, 2005; Brunke et al., 2008). The addition of complexity to any coupling solution must be carefully thought as these cannot become intricate to the point of computation becoming unbearable for ESM application. Algorithms making extensive use of for-loops are unviable: (i) loops are computationally slow and (ii) can easily become conflictive with vectorization and its coordination with parallel processing. Hence, our software needed a deep restructuration from its original version presented in Vieira et al. (2013). Once done, vectorization *per se* enabled improving calculations roughly $12\times$ faster in a single core.

The COARE algorithm is the state-of-the-art in the estimation of atmosphere-open ocean fluxes. During most of its development it focused on the estimation of the heat and humidity fluxes using a framework with an intricate mathematical structure going deep into the simulation of the geophysical process. Only later did it explicitly address the gas fluxes (Fairall et al, 2003; Blomquist et al., 2006, 2014; Jeffrey et al., 2010). Unfortunately, its complexity is deterrent of application to already computationally intensive geoscientific models. The transferred quantities of heat, moisture and momentum, their transfer coefficients, the dimensionless roughness lengths and the gustiness term are interdependent and must be solved iteratively. The COARE estimation of the $\psi_m$ alone is computationally heavier than the most comprehensive ensemble possible in the FuGas 2.2. The COARE was recently introduced as optional in the MOHID model and software. Our preliminary trials testing the heat transfer took $\approx 14\%$ longer to run each simulation.

**5      Code and Data Availability**

[revised manuscript text omitted]

OCMIP: Ocean Carbon-Cycle Model Intercomparison Project, available at: http://ocmip5.ipsl. jussieu.fr/OCMIP, last updated: 2004 (last access: 27 August 2015), 2004.

Pereira, R., Scheneider-Zapp, K. and Upstill-Goddard, R.: Surfactant control of gas transfer velocity along an off-shore coastal transect: results from a laboratory gas exchange tank, Biogeosciences Discuss., doi:10.5194/bg-2016-7, 2016.

Raymond, P. A. and Cole, J. J.: Gas exchange in rivers and estuaries: choosing a gas transfer velocity, Estuaries, 24, 312–317, 2001.

Rutgersson, A., Norman, M., Schneider, B., Petterson, H. and Sahlée, E.: The annual cycle of carbon dioxide and parameters influencing the air–sea carbon exchange in the Baltic Proper, J. Mar. Sys., 74: 381-394, 2008.

Sander, R.: Compilation of Henry's law constants (version 4.0) for water as solvent, Atmos. Chem. Phys., 15, 4399-4981, doi:10.5194/acp-15-4399-2015, 2015.

Sarmiento, J. L. and Gruber, N.:. *Ocean Biogeochemical Dynamics*. Princeton University Press, New Jersey, USA. pp73-100, 2013.

Shuiqing, L. and Dongliang, Z.: Gas transfer velocity in the presence of wave breaking. Tellus B, 68, 27034, 2016.

Soloviev, A., Donelan, M., Graber, H., Haus, B. and Schlüssel, P.: An approach to estimation of near-surface turbulence and CO2 transfer velocity from remote sensing data, J. Mar. Sys., 66, 182-194, 2007.

Stull, R. B.: An introduction to Boundary Layer Meteorology, Kluwer Academic Publishers, Dordrecht, pp151-195, 1988.

Sweeney, C., Gloor, E., Jacobson, A. R., Key, R. M., McKinley, G., Sarmiento, J. L. and Wanninkhof, R.: Constraining global air-sea gas exchange for CO2 with recent bomb [14]C measurements, Global Biogeochem. Cy., 21, GB2015, doi:10.1029/2006GB002784, 2007.

Takahashi, T., Sutherland, S. C., Sweeney, C., Poisson, A., Metzl, N., Tilbrook, B., Bates, N.,Wanninkhof, R., Feelyf, R. A., Sabine, C., Olafsson, J., and Nojirih, Y.: Global sea–air CO2 flux based on climatological surface ocean $pCO_2$, and seasonal biological and temperature effects, Deep-Sea Res., 49, 1601–1622, 2002.

Taylor, P. K. and Yelland, M. J.: The dependence of sea surface roughness on the height and steepness of the waves, J. Phys. Oceanogr., 31, 572–590. 2001.

Vieira, V. M. N. C. S., Martins, F., Silva, J. and Santos, R.: Numerical tools to estimate the flux of a gas across the air–water interface and assess the heterogeneity of its forcing functions. Ocean Sci., 9, 355-375, 2013.

570     Wang, B., Liao, Q., Fillingham, J. and Bootsma, H. A.: On the coefficients of small eddy and surface divergence models for the air-water gas transfer velocity, J. Geophys. Res. Oceans, 120(3), DOI: 10.1002/2014JC010253, 2015.

Wanninkhof, R.:Relationship between wind speed and gas exchange over the ocean, J. Geophys. Res., 97, 7373–7382, 1992.

575     Wanninkhof, R., Asher, W. E., Ho, D. T., Sweeney, C. S., and McGillis, W. R.: Advances in quantifying air-sea gas exchange and environmental forcing, Ann. Rev. Mar. Sci., 1, 213–244, doi:10.1146/annurev.marine.010908.163742, 2009.

Webb, E. K., Pearman, G. I. and Leuning, R.: Correction of flux measurements for density effects due to heat and water vapour transfer, Quart. J.R. Meteorol. Soc., 106: 85-100, 1980.

580     Weiss, R. F.: Carbon dioxide in water and seawater: the solubility of a non-ideal gas, Mar. Chem., 2, 203-215, 1974.

Weiss, R. F. and Price B. A.: Nitrous oxide solubility in water and seawater, Mar. Chem., 8, 347-359, 1980.

Woolf, D. K.: Parameterization of gas transfer velocities and sea state-dependent wave breaking,Tellus B, 57, 87 – 94, 2005.

Woolf, D. K., Leifer, I. Nightingale, P.D., Andreae, M.O.: Modelling of bubble-mediated gas transfer: Fundamental
585     principles and a laboratory test, J. Mar. Sys., doi: 10.1016/j.jmarsys.2006.02.011, 2007.

Wu, L., Rutgersson, A. and Sahlée, E.: The impact of waves and sea-spray on modelling storm track and development, Tellus A, 67, 27967, 2015.

Zeng, X. and Beljaars, A.: A prognostic scheme of sea surface skin temperature for modelling and data assimilation. Geophys. Res. Lett., 32, L14605, 2005.

590     Zhang, W., Perrie, W. and Vagle, S.: Impacts of winter storms on air-sea gas exchange. Geophys. Res. Lett., 33, L14803, 2006.

Zhao, D., Toba, Y., Suzuki, Y., and Komori, S.: Effect of wind waves on air-sea gas exchange: proposal of an overall $CO_2$ transfer velocity formula as a function of breaking-wave parameter, Tellus B, 55, 478–487, 2003.

Zhao, D., Toba, Y., Sugioka, K. and Komori, S.: New sea spray generation function for spume droplets. J. Geophys.
595     Res., 111(C2), doi: 10.1029/2005JC002960, 2006.

600

605

**8 Figures**

[Figure]

**Figure 1:** **Comparing solubility formulations**

[Figure]

**Figure 2: Bias in the gas mass balance for the European coastal ocean**

[Figure]

**Figure 3**: **Comparing transfer velocity algorithms using the data observed at the Baltic.**

[Figure]

**Figure 4**: **Elasticities of the transfer velocity to the environmental variables**.

[Figure]

**Figure 5: Applying the modelled data about the European coastal ocean for a direct comparison between the $k_w$ estimates**.

[Figure]

**Figure 6: Comparing transfer velocity algorithms using modelled data**

**9        Figure legends**

**Figure 1: Comparing solubility formulations**: match-mismatch between formulations estimating solubilities of the greenhouse gases $CO_2$, $CH_4$ and $N_2O$ according to water temperature ($T_w$), salinity (S) and location. Colorscale is quotient between $k_H$ estimated from Johnson, 2010 ( $k_{H\cdot Joh10}$) and $k_H$ estimated from Sarmiento and Gruber, 2013 ($k_{H\ 'Sar13'}$) i.e, $k_{H'Joh10'}/k_{H\ 'Sar13'}$.

**Figure 2: Bias in the gas mass balance for the European coastal ocean**: comparing algorithm by Johnson (2010) to compilation by Sarmiento and Gruber (2013). Colorscale: $\Delta$ton $\cdot m^{-1}\cdot 121$ $km^{-2}$ i.e, bias in the gas mass estimated by each algorithm ($\Delta$ton) for the first meter depth ($m^{-1}$) in 11 km wide cells (121 $km^{-2}$).

**Figure 3: Comparing transfer velocity algorithms using the data observed at the Baltic**. (a) sea-surface roughness given by significant wave height ($H_s$) and peak wave period ($L_p$). (b) the $k_w$ estimated by $u_{10}$-based formulations (lines)and compared to the $k_w$ estimated from the Eddy-Covariance measurements (circles). (c) comparing the $k_w$ estimated by $u_{10}$-based formulations (lines) and by comprehensive alternatives (circles). Simple formulations by 'Wan92' - Wannninkhof (1992), 'WMG99' – Wanninkhof and McGillis (1999), 'We09' – Wanninkhof et al. (2009), 'Sea07' – Sweeney et al. (2007), 'Nea00' – Nigthingale et al. (2000), 'McG01' – McGillis et al. (2001), 'Ho+06' – upper boundary in Ho et al. (2006) 'Ho-06' – lower boundary in Ho et al. (2006) . Comprehensive formulations were assembled using the 'iWLP' – iteratively estimated wind log-linear profile and included the 'Jea87' – Jähne et al (1987), 'ZRb03' -  Zhao et al (2003), and 'W05av' – Woolf (2005) with the kinematic viscosity of air.

**Figure 4: Elasticities of the transfer velocity to the environmental variables.**  Elasticities $(\partial k_w/k_w)/(\partial x/x)$ estimated using the data observed at the Baltic. The $k_w$ was estimated by the iterative wind log-linear profile (iWLP) with the Zhao et al (2003) $k_{bubble}$ term (ZRb03) for the 60 observations in the Baltic. The box-and-wiskers represent the quartiles.

**Figure 5: Applying the modelled data about the European coastal ocean for a direct comparison between the $k_w$ estimates** by the ESM standard –Wan92 - and a comprehensive formulation - W05va - including the $k_{bubble}$ term by Woolf (2005), the $k_{wind}$ term by Jahne et al. (1987), the $z_0$ term from the COARE 3.0 and the iterative wind log-linear profile. The $z_0$ is given in m and the $u_*$ in $m\cdot s^{-1}$.

**Figure 6: Comparing transfer velocity algorithms using modelled data**: ($k_w$ $CO_2$ Wan92) transfer velocity of $CO_2$ estimated from the formulation by Wanninkhof (1992) and averaged over the 66 h; (RMSE) Root Mean Square Error between estimating the transfer velocity using the formulation by Wanninkhof (1992) or the formulation by Woolf (2005) conjugated with the iterative wind log-linear profile.

---

## Author Comment (AC5) · 10 Jan 2017

Several upgrades were made to the FuGas 2.2, namely:

1. Were added the kw algorithms by Ho et al. (2006) and Wanninkhof et al. (2009).

2. Were added the COARE equations for the atmospheric stability, although not yet iterated for convergence.

3. The data management was improved to decrease significantly the Ram memory consumption.

4. The estimation of u10 from the wind log-linear profile was automated.

5. Were added some comments to the code with warnings and instructions.